# Novel travel time aware metapopulation models and multi-layer waning immunity for late-phase epidemic and endemic scenarios

**Henrik Zunker**[1], **René Schmieding**[2], **David Kerkmann**[2], **Alain Schengen**[3], **Sophie Diexer**[4], **Rafael Mikolajczyk**[4], **Michael Meyer-Hermann**[2], **Martin J. Kühn**[1,5]*

**1** Institute of Software Technology, Department of High-Performance Computing, German Aerospace Center, Cologne, Germany, **2** Helmholtz Centre for Infection Research, Brunswick, Germany, **3** Institute for Transport Research, German Aerospace Center, Cologne, Germany, **4** Institute for Medical Epidemiology, Biometrics and Informatics, Medical Faculty of the Martin Luther University Halle-Wittenberg, Halle, Germany, **5** Life and Medical Sciences Institute and Bonn Center for Mathematical Life Sciences, University of Bonn

* Martin.Kuehn@dlr.de

**Data Availability Statement:** The code is publicly available on GitHub under https://github.com/SciCompMod/memilio. The OD matrices used in

## Abstract

In the realm of infectious disease control, accurate modeling of the transmission dynamics is pivotal. As human mobility and commuting patterns are key components of communicable disease spread, we introduce a novel travel time aware metapopulation model. Our model aims to enhance estimations of disease transmission. By providing more reliable assessments on the efficacy of interventions, curtailing personal rights or human mobility behavior through interventions can be minimized. The proposed model is an advancement over traditional compartmental models, integrating explicit transmission on travel and commute, a factor available in agent-based models but often neglected with metapopulation models. Our approach employs a multi-edge graph ODE-based (Graph-ODE) model, which represents the intricate interplay between mobility and disease spread. This granular modeling is particularly important when assessing the dynamics in densely connected urban areas or when heterogeneous structures across entire countries have to be assessed. The given approach can be coupled with any kind of ODE-based model. In addition, we propose a novel multi-layer waning immunity model that integrates waning of different paces for protection against mild and severe courses of the disease. As this is of particular interest for late-phase epidemic or endemic scenarios, we consider the late-phase of SARS-CoV-2 in Germany. The results of this work show that accounting for resolved mobility significantly influences the pattern of outbreaks. The improved model provides a refined tool for predicting outbreak trajectories and evaluating intervention strategies in relation to mobility by allowing us to assess the transmission that result on traveling. The insights derived from this model can serve as a basis for decisions on the implementation or suspension of interventions, such as mandatory masks on public transportation. Eventually, our model contributes to maintaining mobility as a social good while reducing exuberant disease dynamics potentially driven by travel activities.

this study have been published freely in https://mobilithek.info/offers/573360269906817024. Contact data is available in the repository itself. All simulation results can be reproduced with the provided data and code repository. The code and the different models are available on different branches. The introduced SECIRS-type model with waning immunity can be found in branch 323-waning-immunity-model. The novel mobility model is available in branch 385-mobility-scheme-with-transmission-and-traveltimes. The combination of both branches and the simulations presented in this paper are available on branch paper-zunker-simulations-2024.

**Funding:** The authors HZ, RS, AS, and MJK have received funding by the German Federal Ministry for Digital and Transport under grant agreement FKZ19F2211A, RS has received funding by the German Federal Ministry for Digital and Transport under grant agreement FKZ19F2211B. HZ, DK, MMH, and MJK have received funding from the Initiative and Networking Fund of the Helmholtz Association (grant agreement number KA1-Co-08). The funders had no role in study design, data collection and analysis, decision to publish, or preparation of the manuscript.

**Competing interests:** The authors have declared that no competing interests exist.

## Author summary

As human contacts and contact networks are key to the development and prediction of infectious disease spread, travel and commuting activities are important components to be considered in mathematical-epidemiological modeling. Two, often contrasting modeling approaches, based on subpopulations and based on individuals can provide insights of different granularity but also come at different levels of complexity. With this article, we extend a recently introduced Graph-ODE-based model by the explicit introduction of mobility-based infection models in which we allow focused nonpharmaceutical interventions, like face mask mandates in public transport, and in which we can explicitly keep track of secondary cases induced by travel activities, a component mostly not available with equation-based models. In addition, we introduce a novel multi-layer waning immunity model particularly suitable for late-phase epidemic or endemic scenarios. On a daily level and geographically small scale, the newly proposed model often develops similarly, although our results show that complex mobility networks can lead to substantially different disease dynamics in the entirety of a federal state or country. The proposed model thus enables a better understanding of infectious disease dynamics through mobility. It allows for targeted numerical investigations and thus leads to more appropriate real-world interventions.

## Introduction

According to the World Health Organization (WHO), the COVID-19 pandemic resulted in 14.9 million excess deaths in 2020 and 2021. Despite ongoing efforts to tackle infectious diseases, WHO predictions foresee communicable diseases to still account for 6% of global deaths by 2048 [1]. The unprecedented development and administration of COVID-19 vaccines has undoubtedly saved many lives and years of live [2]. Past experiences have illustrated that waning immunity [2] and reduced vaccine protection against simple transmission (i.e., any infection) may require continuous pairing of vaccination with nonpharmaceutical interventions (NPIs) such as face masks or physical distancing, in particular in situations of high transmission [3]. For instance, after several vaccination campaigns, most SARS-CoV-2 related interventions had been discontinued in Germany until summer or early autumn 2022. To further protect vulnerable groups, FFP2 masks became mandatory in all long-distance trains on October 1st, 2022 and federal states could implement additional mask mandates in local or regional public transport [4].

In order to proactively react to infectious disease propagation, mathematical models are of great aid. Models of different types have been largely used to guide policy-makers towards evidence-based decisions, see, e.g. [5] for autumn and winter scenarios in Germany in 2022/2023. To increase their impact, models should be combined with automated pipelines [6] and user-centric visual analytics tools [7] to allow for swift and optimized reaction. Over the last years, contributions have been made by a variety of different approaches, to predict SARS-CoV-2 development in Germany. Among these approaches are agent-based and individual-based models [8–12], models based on ordinary differential equations (ODE) [13–15], delay differential equations [16], integro-differential equation-based models [17], advanced ODE-based models [18] using the linear chain trick [19, 20] to waive the implicit assumption of exponentially distributed transition times, hybrid agent-metapopulation models [21], and in particular standard [22–25] or graph-ODE [26] metapopulation models. From the international

community, an even ampler set of models has been suggested to mitigate the spread of SARS-CoV-2. Some highly recognized international models can be found in [10, 11, 27, 28].

Mathematical models based on systems of ODEs, often denoted compartmental models, are popular tools used in the context of modeling infectious diseases, see, e.g., [29]. This is due to their well-understood character and established methods for model analysis but also because of the low computational cost. However, transmission dynamics of communicable diseases naturally follow the complex network structures of human mobility. Standard ODE-based metapopulation (see, e.g., [22–25]) or Graph-ODE models [30] can be seen as a simple compromise between too simple ODE-based and complex agent-based models. They combine the low computational effort of ODE-based modeling on small- or medium-sized regional entities with realistic mobility networks between these regions.

Although we see certain advantages of Graph-ODE models [30] over classic ODE-based metapopulation approaches, the model in [30] neglected travel times and explicit transmission during commuting. In the current paper, we propose an extended travel time-aware Graph-ODE model that allows for explicit modeling of travel time and transmission on travel and commute.

Aside from mobility, disease dynamics can be driven or mitigated by non-protection or protection against a given pathogen. In particular for late-phase epidemic or endemic scenarios, waning immunity is an important factor to be considered. However, a recent review study [31] on models used in different forecast and scenarios hubs showed that only four out of 90 models provided information or modeling on waning immunity. We thus propose a multi-layer waning immunity model based on ordinary differential equations. In this model, we can consider three different subpopulations with corresponding immunity layers and protection factors. Furthermore, protection against mild and severe courses of the disease wane with different paces.

The current paper is organized as follows. In the first part of materials and methods, we introduce a simple ODE-SIR model and extend it by a mobility-model. We then extend a recently proposed hybrid Graph-ODE ansatz to use our novel travel-time aware model locally. Then, we additionally introduce an advanced ODE-SECIRS model containing three different immunity layers and two different paces for waning immunity, against any and severe infection, respectively. Then, we provide parameters for mobility, contact patterns, and transmission dynamics. In the results section, we show how the new model advances the previous one. Eventually, we discuss advantages and limitations and draw a conclusion.

## Materials and methods

In this section, we will introduce a novel travel time aware metapopulation model which will be implemented in a multi-edge graph based on [30] to account for spatially heterogeneous disease dynamics. For a county-level resolution of Germany, we provide travel and commute patterns based on [32]. Furthermore, we introduce a multi-layer waning immunity model based on ordinary differential equations (ODEs) and which is of particular interest in late-phase epidemic or endemic scenarios. Finally, we will summarize parameters used in our estimations for the late-phase SARS-CoV-2 demonstrator case.

### A novel travel time aware metapopulation model

For compartment-based modeling, we need to define a list of *disease* or *infection states*. In the simplest SIR case, we have the states *Susceptible* for persons who can get infected, *Infected* or *Infectious* for persons who can infect others, and *Recovered* or *Removed* for individuals that cannot get reinfected. In order to introduce our travel time aware metapopulation model, we

start from this simple ODE-SIR model which writes

$$
\begin{aligned}
S^{(k)'}(t) &= -\lambda^{(k)}(t)S^{(k)}(t) \\[1.5em]
I^{(k)'}(t) &= \lambda^{(k)}(t)S^{(k)}(t) - \frac{1}{T_I^{(k)}}I^{(k)}(t) \\[1.5em]
R^{(k)'}(t) &= \frac{1}{T_I^{(k)}}I^{(k)}(t) \\[1.5em]
\lambda^{(k)}(t) &= \rho^{(k)}(t)\phi^{(k)}(t)\frac{I^{(k)}(t)}{N^{(k)}(t)}
\end{aligned}
\tag{1}
$$

and where $\phi^{(k)}(t)$ represents the (mean) daily contacts of a person at time t, $\rho^{(k)}(t)$ the (mean) transmission risk, $T_I^{(k)}$ the (mean) time a person stays infected, and $N^{(k)} = N^{(k)}(t) = S^{(k)}(t) + I^{(k)}(t) + R^{(k)}(t)$ the (constant) total population size. Note that in these model, we always use mean or averaged values so that we drop the *mean* or *average* description to parameters in the following.

For the sake of a consistent development of the new model, we have already introduced the index $k$ that will later be used to refer to a particular region $k$ but which is without meaning for the presentation in this subsection.

We suppose some initial conditions at $t_0$ to be given by $(S^{(k)}(t_0), I^{(k)}(t_0), R^{(k)}(t_0))$.

In infectious disease modeling, the number of daily contacts is often estimated from surveys or contact diaries like [33]. These surveys provide us with average contact information $\phi^{(k)}$, summed for all contact locations, and, in particular, traffic-related contacts $\phi_{tr}^{(k)}$. In our model, we allow for nonpharmaceutical interventions that change the daily number of contacts over time. Consequently, we write $\phi^{(k)}(t) := \omega^{(k)}(t)\phi^{(k)}$ and $\phi_{tr}^{(k)}(t) := \omega_{tr}^{(k)}(t)\phi_{tr}^{(k)}$, where $\omega^{(k)}(t)$ and $\omega_{tr}^{(k)}(t)$ are suitable reduction factors of the registered baseline contacts.

In the following paragraphs, we will introduce our mobility-model approach for ODE-based models.

**Classic traffic contacts integration and traffic-related NPIs.** A straightforward approach modeling (local) traffic-related contacts in compartmental models is given with a separation of contacts in traffic and nontraffic. We can then write

$$
\phi^{(k)}(t) = \phi_{tr}^{(k)}(t) + \phi_{nt}^{(k)}(t)
\tag{2}
$$

where $\phi_{nt}^{(k)}(t)$ represents all contacts that happen elsewhere, i.e., in home, school, workplace, or leisure activity settings. If Eq (2) is plugged into Eq (1), we can model a reduction in travel contacts from day $t_1 > t_0$ with factor $r \in [0, 1]$ by using

$$
\widetilde{\phi}_{tr}^{(k)}(t) := \begin{cases}
\phi_{tr}^{(k)}(t), & t \leq t_1 \\[1em]
\widehat{\phi}(t), & t \in (t_1, t_1 + \delta), \quad 0 < \delta < 1, \\[1em]
(1-r)\phi_{tr}^{(k)}(t-\delta), & t \geq t_1 + \delta
\end{cases}
\tag{3}
$$

where we define $\delta$ as a transition interval and $\widehat{\phi}$ as a transition contact rate between the change of the contact rate to ensure that $\widetilde{\phi}_{tr}^{(k)}(t) \in \mathcal{C}^1((t_1 - \epsilon, t_1 + \delta + \epsilon))$ with $0 < \epsilon \ll 1$ for Eq (1) to be well defined.

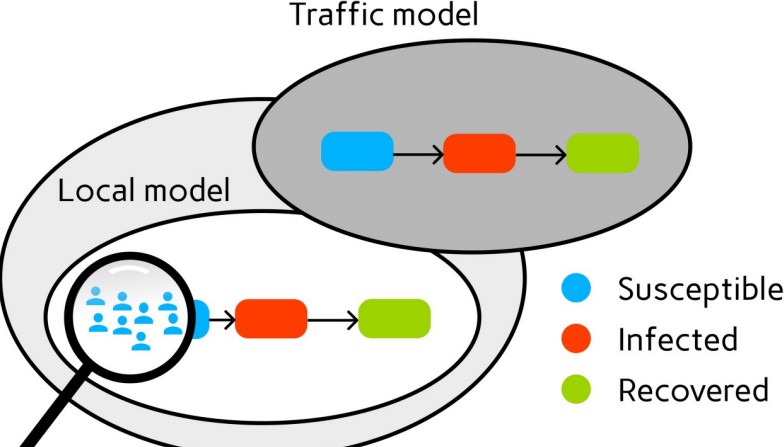

**Fig 1. Compartmental transmission model (white) complemented by compartmental transmission mobility model (gray).**

This type of contact reduction for contact locations such as *home*, *school*, *work*, and *other* has been considered in [26, 30, 34]. For noncontinuously differentiable parameters, we actually compute the solution of a new initial value problems (IVP) from $t_1$.

**Explicit traffic or mobility modeling.**   In order to separate disease dynamics on travel and commute from nontraffic (i.e., home or work) dynamics, we propose a new way by introducing a mobility-based infection model in which dynamics follow the same mechanisms but in which the population distribution as well as transmission or mitigation parameters differ; see Fig 1. In the following and for the sake of simplicity, we will refer to the mobility-based infection model simply as *mobility model* or *traffic model*. Note that this model basically is a pure infection dynamics model but is specifically parametrized to represent mobility settings.

Note that the novel model is indeed different and expected to behave different where it relaxes assumptions such as homogeneous mixture. However, for meaningful comparisons and in order to be consistent, contact patterns across both models have to be adjusted. In the following, we will explain how to scale contacts that are obtained from classical diary studies to our new model.

Without loss of generality, let us assume that all individuals have contacts in nontraffic locations (which is trivial, i.e., in home or work). However, only a subset $p_{tr}^{(k)} \in [0, 1]$ of individuals also have contacts in traffic locations, as some people may not commute at all or may use private transportation modes such as cars without additional passengers such that they do not come into contact with others (on commuting itself). For demonstration purposes, we consider the classic and the new model on a timescale of one day, where each day has unit length, and neglect the daily return trip for now. In the final implementation, this approach is then just down-scaled to half-a-day.

As we are still working with a mean-value model, we assume that mobility completely happens either at the beginning or the end of the day, respectively. We will provide graphs for both realizations, but provide the following description with mobility taking place at the end of the day. To that end, we introduce the daily travel time $0 \ll t_{tr}^{(k)} < 1$, such that the mobility process starts near the end of the day. Note that we will later use a time scale of half a day, with a *length of stay* in another region and *return trip* to the home region, once we introduced a second spatial region.

We further assume that the number of contacts in nontraffic locations, $\phi_{nt}^{(k)}(t)$, is independent of the additional contacts during travel, e.g., the sole fact that a person commutes or uses public transport does (on average) not reduce the number of contacts to family members, friends, or colleagues. In order to compute the number of contacts in traffic modes per individual traveler, we compute

$$\phi_{tr|tr}^{(k)}(t) := \frac{N^{(k)}\phi_{tr}^{(k)}(t)}{p_{tr}^{(k)}N^{(k)}} = \frac{\phi_{tr}^{(k)}(t)}{p_{tr}^{(k)}}. \tag{4}$$

The end-of-day approximation of our novel model is then given after solving three initial value problems. We solve

$$\text{Eq (1)} \quad \text{with} \quad \phi^{(k)}(t) := (1 - p_{tr}^{(k)})\phi_{nt}^{(k)}(t) + p_{tr}^{(k)}\frac{\phi_{nt}^{(k)}(t)}{t_{tr}^{(k)}}, \quad t \in [0, t_{tr}^{(k)}], \tag{5}$$

with initial values $(S_0, I_0, R_0) := (S^{(k)}(0), I^{(k)}(0), R^{(k)}(0))$. The scaling of contacts comes from the fact that $p_{tr}^{(k)}N^{(k)}$ many individuals now have all their nontraffic related contacts within $t_{tr}^{(k)}$ many days, $t_{tr}^{(k)} < 1$. This means that the value $\phi_{nt}^{(k)}(t)$, given on daily scale, needs to be rescaled. Let $\widehat{y}(t_{tr}^{(k)}) := (\widehat{S}^{(k)}(t_{tr}^{(k)}), \widehat{I}^{(k)}(t_{tr}^{(k)}), \widehat{R}^{(k)}(t_{tr}^{(k)}))$ be the solution of (5) at $t_{tr}^{(k)}$. We define the IVP

$$\text{Eq (1)} \quad \text{with} \quad \phi^{(k)}(t) := \phi_{nt}^{(k)}(t), \quad t \in [t_{tr}^{(k)}, 1], \tag{6}$$

with initial values $(1 - p_{tr}^{(k)})\widehat{y}(t_{tr}^{(k)})$ and a total population of $(1 - p_{tr}^{(k)})N^{(k)}$, i.e., the people that do not travel, in the denominator. Furthermore, we define the IVP in the travel node by

$$\text{Eq (1)} \quad \text{with} \quad \phi^{(k)}(t) := \frac{\phi_{tr|tr}^{(k)}(t)}{1 - t_{tr}^{(k)}}, \quad t \in [t_{tr}^{(k)}, 1], \tag{7}$$

with initial values $p_{tr}^{(k)}\widehat{y}(t_{tr}^{(k)})$ and a total population of $p_{tr}^{(k)}N^{(k)}$ in the denominator. The complete solution at $t = 1$ is then given by the summed solutions of (6) and (7). Geometrically, IVPs (5) and (6) are assigned to the white local model from Fig 1, while (7) corresponds to the grey traffic or mobility model.

Here, with $p_{tr}^{(k)}$ and $1 - p_{tr}^{(k)}$, we used an equal and potentially unrealistic distribution of the population compartments to traffic and nontraffic node, respectively. In a more complex model, one would also reduce the relative share of infected commuters, letting $p_{tr}^{(k)}$ susceptible but only $\zeta p_{tr}^{(k)}$, $\zeta \in [0, 1)$, infected individuals commute. This reduction has already been used in [30] and naturally translates to our new model.

In Fig 2, we provide two examples of our new model for the parameters $\phi_{nt}^{(k)} = 9$, $\phi_{tr}^{(k)} = 1$, $p_{tr}^{(k)} = 0.1$, $T_I^{(k)} = 8$, $I_0^{(k)} = R_0^{(k)} = 2$, $N_0^{(k)} = 10000$, $t_{tr}^{(k)} = 0.9$ and $\rho(t) = 0.05$ (left) and $\rho(t) = 0.1$ (right). In the new model, even if the total number of contacts stays the same, the contact structure changes from an average of $\phi$ to a bimodal contact distribution. We see that, both, the mobility-first and mobility-last approach end up with roughly the same number of infections at the end of the day but with, of course, slightly different numbers of susceptible and recovered individuals. We demonstrate the application of the scaling procedure to real contact data in the section *Mobility and contact patterns*.

In the end, model (1) can be substituted with any type of (complex) ODE-based model (not yet integrating mobility processes). However, the modeling of the transmission process can be very different. In other models, a single parameter, often referred to as $\beta$, is used to represent the transmission dynamics [29, 35, 36] and [22–25] directly integrate transmission on or after

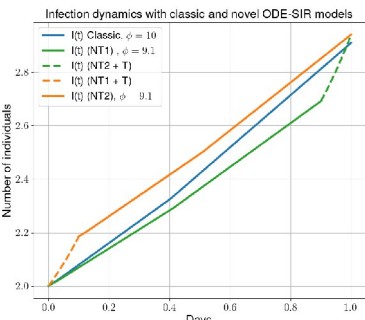
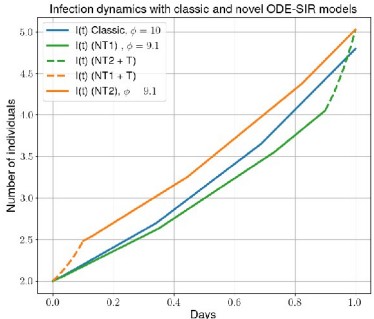

**Fig 2. Infection curves for simple (blue) and mobility-based infection model (orange and green) approach.** The orange curve shows the dynamics with mobility first while the green curve shows the mobility last approach. The example's parameters are $\phi_{nt}^{(k)} = 9$, $\phi_{tr}^{(k)} = 1$, $p_{tr}^{(k)} = 0.1$, $T_I = 8$, $I_0^{(k)} = R_0^{(k)} = 2$, $N_0^{(k)} = 10000$, $t_{tr}^{(k)} = 0.9$ and $\rho(t) = 0.05$ (left) and $\rho(t) = 0.1$ (right). In the legend of the plots, we abbreviate nontraffic with NT and traffic with T.

commuting in the force of infection. In the given presentation, we split this parameter into two different factors, the transmission probability per contact $\rho$ and the contact rate $\phi$. This decomposition allows for a more detailed and flexible parametrization of the transmission process. On top of this splitting, we add a second model realizing transmission on commuting and travel. This additional model can be used, independent of the underlying transmission dynamics model. We admit that by this splitting, seasonal mobility effects are not yet taken into account. However, seasonality is integrated by a parametric trigonometric curve as first described in [30]. This means that seasonality can also be interpreted as a change in the effective contact rate over time. For more details also see (16).

**Theoretical properties of the novel mobility model.** In order to give the motivation for our scaling, we assume constant contact patterns over the considered day, i.e., $\phi_{nt}^{(k)}(t) = \phi_{nt}^{(k)}$ and $\phi_{tr}^{(k)}(t) = \phi_{tr}^{(k)}$, $t \in [0, 1]$. For the total number of contacts, we then directly obtain

$$
\int_0^{t_{tr}^{(k)}} N^{(k)} \left( (1 - p_{tr}^{(k)}) \phi_{nt}^{(k)} + p_{tr}^{(k)} \frac{\phi_{nt}^{(k)}}{t_{tr}^{(k)}} \right) dt + \int_{t_{tr}^{(k)}}^1 (1 - p_{tr}^{(k)}) N^{(k)} \phi_{nt}^{(k)} dt
$$

$$
+ \int_{t_{tr}^{(k)}}^1 p_{tr}^{(k)} N^{(k)} \frac{\phi_{tr|tr}^{(k)}}{1 - t_{tr}^{(k)}} dt \tag{8}
$$

$$
= N^{(k)} \left( (1 - p_{tr}^{(k)}) \phi_{nt}^{(k)} t_{tr}^{(k)} + p_{tr}^{(k)} \phi_{nt}^{(k)} + (1 - p_{tr}^{(k)}) \phi_{nt}^{(k)} (1 - t_{tr}^{(k)}) + \phi_{tr}^{(k)} \right)
$$

$$
= N^{(k)} \left( \phi_{nt}^{(k)} + \phi_{tr}^{(k)} \right),
$$

where we recognize Eq (2).

Let us note that the parameters of baseline contacts in traffic $\phi_{tr}^{(k)}$ as well as the travel time $t_{tr}^{(k)}$ that implicitly appear in (7) need to fit together, e.g., coming from reliable data sources of the same country. That means that the model outcome for, e.g., fixed $\phi_{tr}^{(k)}$ and arbitrary $t_{tr}^{(k)}$ is not anymore in line with the data used in the model. However, in the following sections, we still allow a certain variance in travel time between different regions, i.e., $1 - t_{tr}^{\max} < 1 - t_{tr}^{(k)} < 1 - t_{tr}^{\min}$, $k \in \{1, \dots, n\}$. Due to data sparsity, i.e., no individual traffic contact records for arbitrary commuting routes ($\phi_{tr}^{(k)} = \phi_{tr}$, $k \in \{1, \dots, n\}$), we only assume

**Fig 3. Hybrid graph-ODE approach [30] (left) and novel extension with mobility models (right).** For simplification, we visualized an SIS model with two sociodemographic groups (e.g. age groups). In the old approach (left), no travel time was used while the novel approach models (right) travel time through compartmental mobility models.

that

$$\frac{\phi_{tr}}{1 - t_{tr}^{\max}} := \text{const} \tag{9}$$

to consider the limit case without travel times, instead of directly considering (7).

Considering the limit $t_{tr}^{(k)} \to 1$ implies $\phi_{tr}^{(k)} \to 0$ (with the same convergence rate). From (2), we have

$$\phi^{(k)}(t) \to \phi_{nt}^{(k)}(t) \tag{10}$$

and (5) also reduces to (10) on $t \in [0, 1]$. The corresponding IVPs (6) and (7) can be dropped as the interval converges to a null set and the parameters remain bounded.

Thus, we see that the novel model advances the previous model [30] by removing the unrealistic assumption of instantaneous travel and allowing for realistic travel times $1 - t_{tr}^{(k)}$.

**Mobility complemented metapopulation models through a multi-edge graph.** In order to make use of the novel model from the previous section, assume $n$ different geographic units (here, denote *regions*) to be given. We now combine our novel mobility model extension of a simple ODE model with the graph approach proposed in [30]. For the paper to be self-contained, we visualize the previous approach in Fig 3 (left). There, each region is represented by the node of a graph while the (multi-)edge $\mathcal{E}_{ij}$ between nodes $\mathcal{N}_i$ and $\mathcal{N}_j$ represents the (outgoing) mobility and mobility-based exchange is realized instantly twice a day (round trip). In practice, each combination of sociodemographic group $g_l$, $l = 1, \ldots, G$ and infection state $z_l$, $l = 1, \ldots, Z$ can be assigned a number of travelers such that the multi-edge consists of $G \times Z$ single edges. Return trips are realized by a mapping on the same edge, as the vector of weights on $\mathcal{E}_{ij}$ represents the number of outgoing travelers. For the sake of a simple visualization in Fig 3 (left), each region was assigned an ODE-SIS model with only two compartments ($S$ and $I$) and two (sociodemographic) groups (e.g., age groups).

This approach extends classic ODE-based models but simplifies the mobility pattern by instantaneously transferring commuters to their intended destinations, omitting the duration of transit. Upon reaching their destinations, individuals remain there for the day's remainder before they immediately return to their origin. In Fig 4 (left), we provide a schematic representation of the mobility distribution under this mobility scheme. This assumption offers the advantage of facilitating the coordination and incorporation of commuters into and out of various regions by standardizing the timing of exchanges. It affords flexibility in selecting larger time steps for the applied numerical integration scheme. However, it also abstracts away

**Fig 4. Timing of the mobility activities.** The hybrid Graph-ODE approach from [30] (left) synchronizes population exchanges at two common daily junctures. The new approach (right) defines an individual structure of mobility activities for each commuter group, which is considering the travel time and the length of stay in the destination. Within this daily schedule, all commuters still return to their home location at the same time, exactly at the end of the day.

critical variables such as the travel time. The instantaneous nature of exchanges precludes the possibility of transmission events during transit, an aspect that can become significant in disease spread modeling.

Deviating from this approach, our novel approach introduces a refined segmentation of the mobility activity schedule by delineating distinct mobility profiles based on local travel and stay times; see Fig 4 (right). The length of stay can be chosen differently for each region. An even more refined selection, e.g., an individual selection for each edge (i.e., a subpopulation of a predefined socioecomic group with a given infection state), would make the scaling of the contacts considerably more complex. This granular approach not only enhances the precision of our method but also mirrors the complex, real-world patterns of human movement more closely. However, including these details means that mobility activities are much more spread out throughout the daily schedule. Therefore, we have to choose substantially smaller time steps in the numerical integration scheme in order to take into account outgoing or incoming commuters.

In the novel approach, traveling is realized via mobility or traffic models; see Fig 3 (right). Again, each region gets assigned a local compartmental model, now complemented by a local compartment mobility model.

A key advantage of augmenting the framework with local compartment mobility models lies in the creation of realistic travel chains. The idea is that commuters pass through other regions during the trip and that the individuals spend time in the mobility models of the region they are passing through. This temporary inclusion enables potential interactions with other commuter groups that are simultaneously represented in the same mobility model.

The trip chain $\mathcal{T}^{(j,k)}$ between two arbitrary nodes $\mathcal{N}_j$ and $\mathcal{N}_k$ is defined as an ordered tuple of regions,

$$\mathcal{T}^{(j,k)} = (\mathcal{N}_j, \mathcal{N}_{R_1}, \mathcal{N}_{R_2}, \ldots, \mathcal{N}_k), \quad R_i \in \{1, \ldots, n\} \setminus \{j, k\}, \tag{11}$$

where $j$ is the index of the starting node, $k$ is the index of the destination node. The transit nodes $\mathcal{N}_{R_1}, \mathcal{N}_{R_2}, \ldots$ describing the way of travel between these regions.

The construction of these travel chains is achieved by calculating the centroids of both the origin and destination regions, represented by (multi-)polygons. A line is then drawn between these centroids; see Fig 5. The sequence in which these regions are encountered along the line is crucial, as it establishes the order of the travel itinerary. In the final step, the overall travel duration is allocated proportionally across the regions in the defined trip chain.

Given that the visualized trip from Region $\mathcal{N}_i$ to Region $\mathcal{N}_k$ in Fig 3 (right) is a direct trip, a traveler spends half of the travel time in each of the corresponding two mobility models. For a trip from Region $\mathcal{N}_i$ to Region $\mathcal{N}_j$, dissected to a trip chain $\mathcal{T}^{(i,j)} = (\mathcal{N}_i, \mathcal{N}_k, \mathcal{N}_j)$, a third of the trip is spent in the single mobility models. This forms our extended hybrid graph-ODE

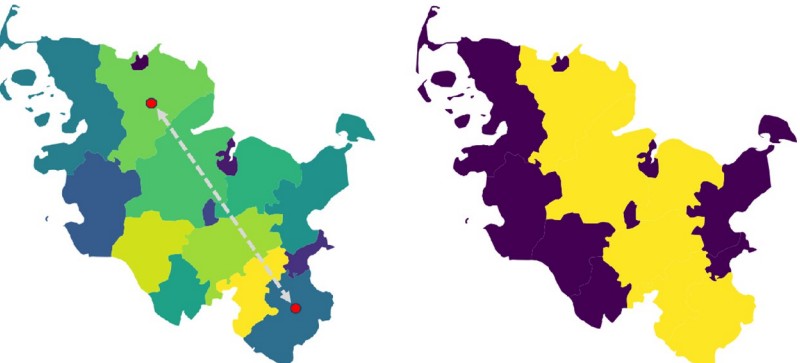

**Fig 5. Determination of the path chains.** Given several regions, we draw a line between the start and destination region (left). Afterwards we consider all regions that are hit by this line and determine the path chain based on the order (right). Using geodata "Verwaltungsgebiete 1:2 500 000, Stand 01.01. (VG2500)" from https://gdz.bkg.bund.de, copyright; GeoBasis-DE / BKG 2021, license dl-de/by-2-0, see https://www.govdata. de/dl-de/by-2-0.

model. The arrows in Fig 3 (right) are representing mobility between the different mobility models.

To accurately model mobility, it is crucial to first identify which individuals are participating in commuting activities. We would like to note here that different travel restrictions can be active for each region, but there should also be the possibility of restrictions on individual edges, e.g., for the subpopulations of detected or symptomatic individuals. We determine the number of commuters in compartment $z \in \mathcal{Z}$ leaving region $\mathcal{N}_j$ for region $\mathcal{N}_k$ by

$$z^{(j,k)}(t) = \frac{z^{(j)}(t)}{N^{(j)}(t)} d_z^{(j,k)}(t) \omega_z^{(j)} C^{(j,k)}, \ z \in \mathcal{Z}, \tag{12}$$

where $\mathcal{Z}$ represents the compound of all compartments of the considered local model. Furthermore, $d^{(j,k)}$ denotes a local reduction factor in daily commuting activity which can be defined for each edge, i.e., any combination of sociodemographic group and infection state. Region-based restrictions, e.g., isolation of particular infection states or general stay home restrictions, can be chosen with $\omega_z^{(j)}$. For example, we can use $\omega_{I_{Sy,*}}^{(j)}$ to isolate symptomatic individuals and to restrict their commuting activities. Finally, we have the total number of people moving between nodes $\mathcal{N}_j$ and $\mathcal{N}_k$, denoted by $C^{(j,k)}$. In summary, we break the total given number of people moving from one region to another, into subpopulations of different infection states for which different restrictions can be applied. These numbers are time dependent.

An important challenge of the approach is tracing the compartmental affiliation of the traveling individuals. We address this issue by employing and extending the method described in [30], where we perform an approximation step using a single-step integration method. With this step, we solely advance the infection states of a focus group of individuals (i.e., travelers) while considering all other persons within the same model (travelers from other destinations and local population) as contact persons only. This approach is independent of the model selected and offers some kind of subpopulation-tracing in a population-level model. As the local travel time is generally small, we also apply a single step method for traffic nodes where only one subpopulation is present. Fig 6 demonstrates the movement patterns of individual groups and how interactions between different groups can lead to additional infections during a trip. Please note that visualization is based on individuals while in practice only subpopulation shares are known.

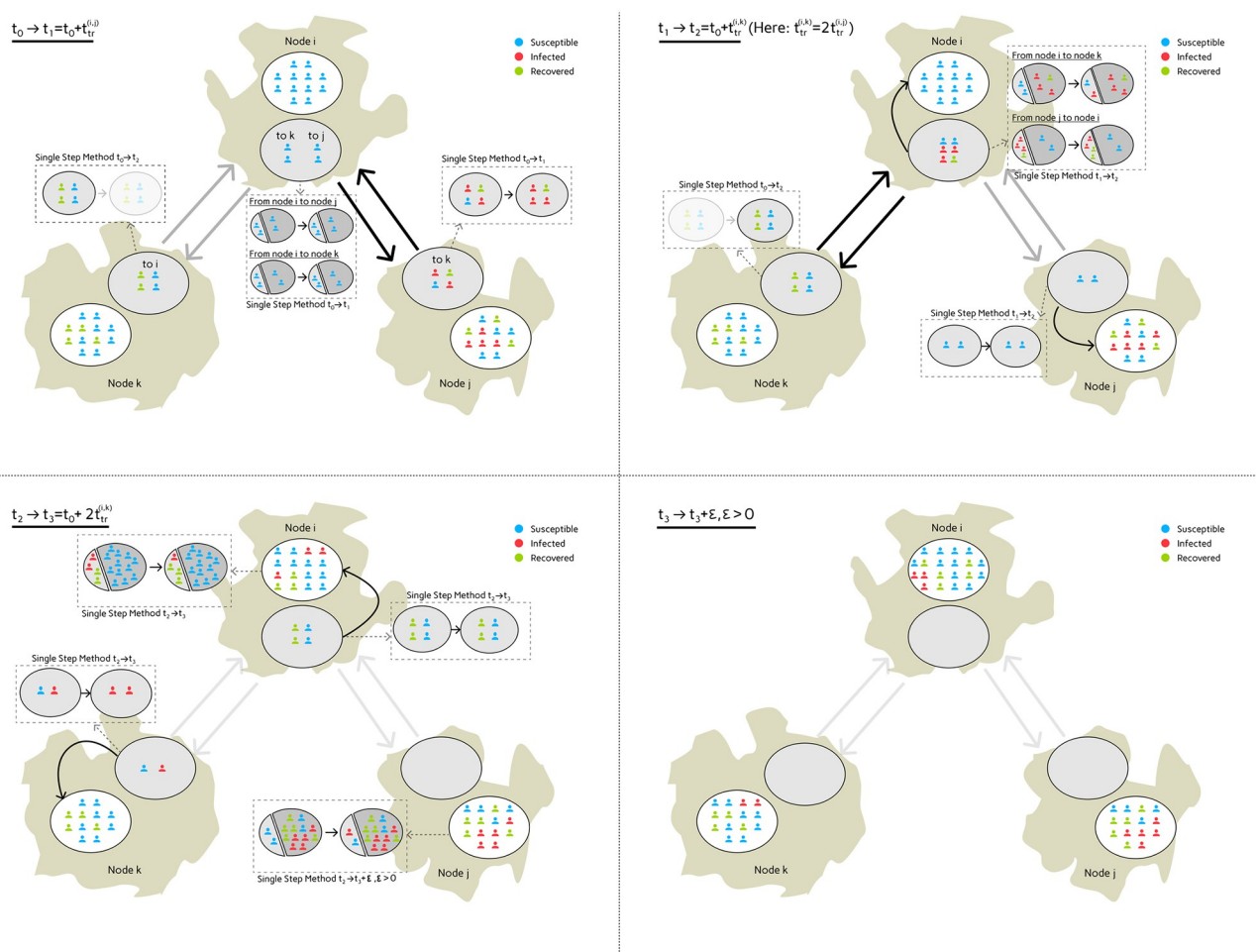

**Fig 6. Demonstration of the mobility across three nodes and four edges.** The travel time between region $i$ and $k$, $t_{tr}^{(i,k)}$, is two times the travel time from region $i$ to $j$, i.e., $t_{tr}^{(i,k)} = 2t_{tr}^{(i,j)}$. For simplification, there is no exchange between nodes $k$ and $j$. The images read from top left to bottom right and all subpictures visualize the developments between (and at the end of) the considered time frame, i.e., $t_0 \to t_1$ for top left. At time $t_0$, for all graph nodes, travelers change from their local infection model to their local traffic infection model. During the interval $[t_0, t_1]$ (top left), state changes in traffic infection models are approximated by a single step time integration method while any other (generally an adaptive high-precision scheme like Cash Karp 5(4) [37]) method can be used in the local nodes. Subpopulations that come from or go to a different location are only considered as contact populations; see [30]. This is the first part of the functionality implemented on the graph's (multi) edges. At $t_1$, the exchange between nodes $i$ and $j$ happens. By $t_1$ (top right), the exchange along the edge between nodes $i$ and $j$ has already happened and three infected plus one recovered individual have moved from travel model of node $\mathcal{N}_j$ to travel model of node $\mathcal{N}_i$ and vice versa. In $[t_1, t_2]$, the individuals present in travel node $\mathcal{N}_i$ and subject to move to $k$ get in contact with the individuals arriving from node $\mathcal{N}_j$. After another time step until $t_2$, the travelers between node $\mathcal{N}_i$ and node $\mathcal{N}_k$ are exchanged. From $t_2$ to $t_3$ (bottom left), we further advance the newly exchanged travelers to finally arrive in the local models at $t_3$, i.e., after they have completed their predefined travel time within the mobility model. For the local models in nodes $i$ and $j$, we must update the states of the arrived individuals using the introduced method, as both local residents and travelers are present, and we will integrate new individuals to the region in the next step. By $t_3 + \varepsilon, \varepsilon > 0$ (bottom right), all individuals have reached their destination regions, i.e. the travel models are empty, and local models can be advanced in parallel.

## A multi-layer waning immunity model of SECIRS-type

Waning immunity is a paramount feature in epidemiological modeling if late-phase epidemic or endemic scenarios are considered and immunity of individuals is not everlasting. This, for instance, is the case for SARS-CoV-2 or influenza, where neither infection nor vaccination establishes lasting protection against (any) re-infection; see, e.g., [38].

To motivate and, later, parametrize our model, we first looked into [38], which considered Influenza with vaccination and reinfections (with new variants) as well as an early result for SARS-CoV-2 [2]. We then also considered two recent systematic reviews with 26 studies in [39] and 65 studies from 19 different countries in [40]. According [39], protection against hospitalisation or serious illness due to previous infections was found to be 74.6% (on average) after 12 months, waned from 80–90% after two months. In individuals with hybrid immunity, i.e., a combination of infection and vaccination, the protective effect of hybrid immunity following primary series remained high at 97.4% after 12 months and 95.3% after 6 months following the first booster vaccination—with practically no waning detected. The paper also shows that protection against reinfection after a previous SARS-CoV-2 infection decreases from roughly 70% after two months to 24.7% within 12 months. For individuals with hybrid immunity with primary series vaccination, protection against reinfection after the first booster vaccination fell from 75–80% to 41.8% within one year interval. Individuals with hybrid immunity due to first booster vaccination had a protection of 46.5% after 6 months. A further systematic review and meta-analysis by [40] found similar results. Protection against (any) reinfection by the Omicron BA.1 variant fell to around 36.1% after 40 weeks, with slightly faster decline against symptomatic reinfection. Protection remained higher for earlier variants but also decreased over time. Again, protection against serious illness stayed high for all variants, including Omicron BA.1, i.e., at 88.9% after 40 weeks. Earlier variants even showed protection of over 90%. These results confirm that despite the decline in protection against reinfection, protection against severe courses lasts longer and remains robust. Overall, we conclude that the immunity and decline in immunity plays a major role in the course of time and should be considered with two different paces.

In the following, we present the full SECIRS-type model with waning immunity; see Fig 7. The motivation for the particular model is threefold and not only in waning immunity. First, as pre- or asymptomatic transmission plays an important role for SARS-CoV-2, we use a differentiation in nonsymptomatic and symptomatic transmission to separate their related contributions to the disease dynamics. Second, we use three different layers of immunity or subpopulations with different protection factors against mild and severe courses of the disease. Finally, we introduce the compartments of *temporary immunity* to realize different paces of immune waning against mild and severe infections, respectively. As motivated above, our model realizes quickly waning protection effects against transmission or any infection while it preserves longer-lasting immunity against severe or critical infections. To account for age-specific transmission and infection parameters, our model will also be stratified for age groups. Our open-source implementation in *MEmilio* [41] allows for a variable number of age groups as well as for flexible integration of other sociodemographic factors such as income.

Our model is based on the previously developed SECIR-type model in [26] which had been assessed as high-quality by a recent meta review [42]. The model allows the integration of vaccinations and different immunity layers but lacks waning immunity. However, the introduction of the *temporary immunity* states changes the meaning of recovery. Recovery always means to enter a temporary immunity compartment and then a susceptible compartment with different protection layer. Furthermore, the states enable the simulation of waning immunity with different paces to suitably model endemic or late-phase epidemic scenarios. The basic infection process of our model follows an *susceptible* ($S$), *exposed* ($E$), *nonsymptomatic infectious* ($I_{NS}$ or $C$ for carrier), and, potentially, *symptomatic infectious* ($I_{Sy}$), *infected severe* ($I_{Sev}$), *infected critical* ($I_{Cr}$), and *Dead* ($D$) structure.

We refer to the first subpopulation (shown in gray in Fig 7) as the *naive* subpopulation. Individuals of this group have either not seen vaccination or infection, did not have a substantial immune reaction to these or immunity has waned completely, with the last infection or

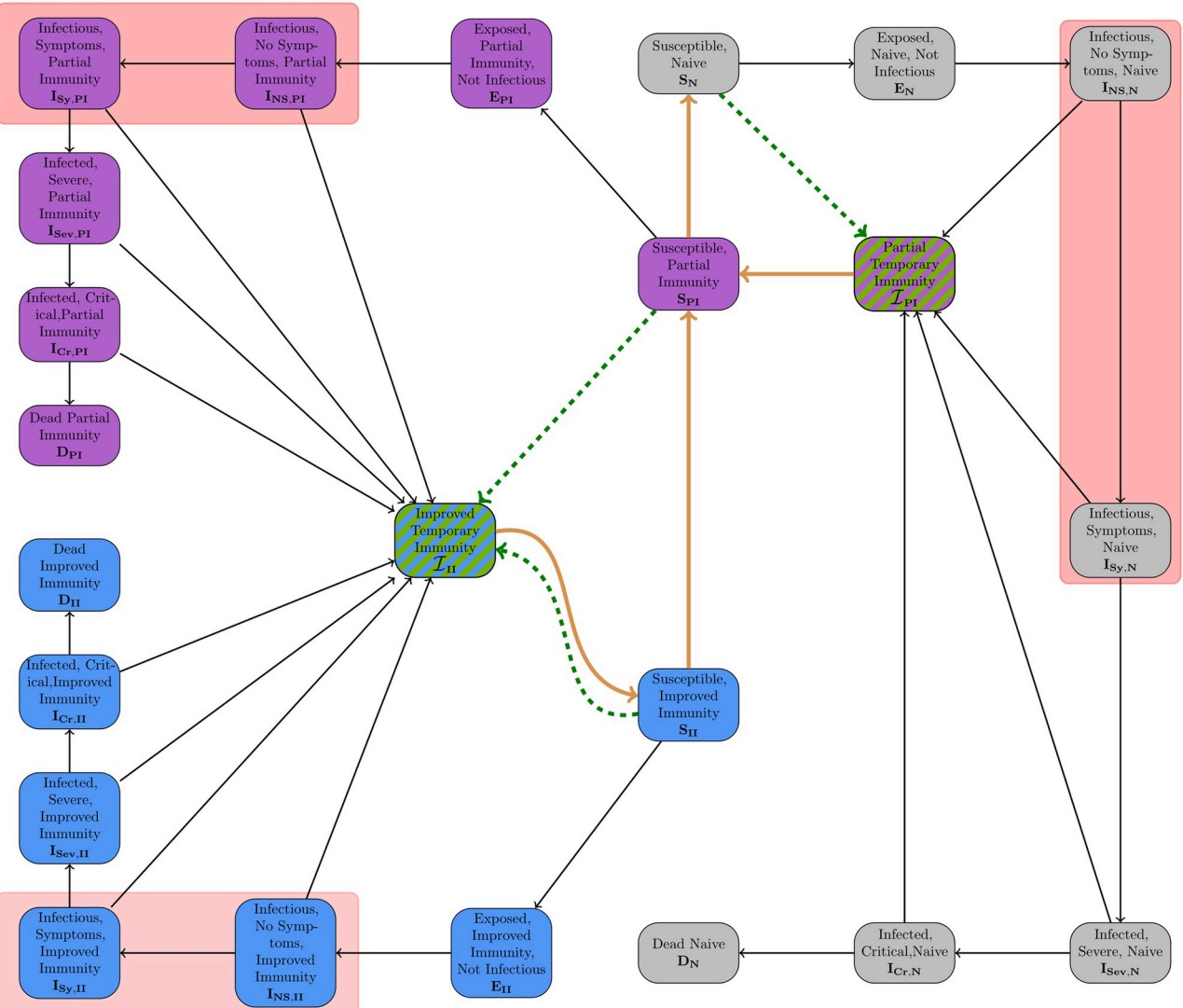

**Fig 7. SECIRS-type model with high temporary immunity upon immunization and waning immunity over time.** The three different immunity layers are visualized by different colors. The waning is represented by orange arrows. Vaccinations are visualized through dashed green arrows and reduce susceptibility for transmission for a certain (short) period of time. Recovery happens from any kind of infection state to a temporary immunity state with normal black arrow. We have highlighted the disease states responsible for transmission by red boxes. For the sake of simplification, we do not visualize the age group model here.

vaccination event long ago. Individuals with *partial immunity* (shown in purple) have an average protection, while individuals with *improved immunity* (shown in blue) are best protected against the pathogen. The arrows in Fig 7 show a permeable model where individuals can change the subpopulation during simulation.

The different states of the model and their meanings, independent of the particular immunity, are shown in Table 1. To further motivate our model development, we have sketched out the idea of different protection layers and waning immunity paces on the individual layer in Fig 8.

In order to cover age group-specific characteristics of the viral dynamics, we stratify the population by different groups $i \in \{1, \ldots, G\}$, where $G$ is the number of age groups. For all parameters and variables, we add the subscript $i$ to distinguish the different age groups.

**Table 1. Model compartments and meanings for disease transmission modeling.**

| Compartment | Symbol | Description |
|---|---|---|
| Susceptible | $S$ | Currently not infected, susceptible to infection. |
| Exposed | $E$ | Infected but not yet infectious. |
| Infected, no symptoms | $I_{NS}$ | Infected and infectious but pre- or asymptomatic. |
| Infected, symptoms | $I_{Sy}$ | Symptomatic, infected and infectious. |
| Infected, severe | $I_{Sev}$ | Treated in hospital, isolated. |
| Infected, critical | $I_{Cr}$ | Treated in intensive care, isolated. |
| Dead | $D$ | Died, removed from model. |
| Temporary immunity | $\mathcal{I}$ | Temporarily immune after completed vaccination or infection. |

In order to introduce the model equations, we define $N_i^{(k),D^\perp}$ as people from age group $i \in \{1, \ldots, G\}$ which are not in the dead compartment. Furthermore, denote by $z_{j,i}^{(k)}$, the $j$-th disease state and by $z_{j+1,i}^{(k)}$ the worsened disease state, e.g., $z_{j,i}^{(k)}$: Infectious, No Symptoms, Naive (age group $i$) and $z_{j+1,i}^{(k)}$: Infectious, Symptoms, Naive (age group $i$). We order the 18 disease states $\mathcal{Z}_I$ as follows: Exposed Naive Not Infectious, ..., Dead Naive, Exposed Partial Immunity Not Infectious, ..., Dead Improved Immunity.

Let $\mu_{z_{j,i}}^{z_{j+1,i},(k)}$ be the probability of transition from disease compartment $z_{j,i}^{(k)}$ to $z_{j+1,i}^{(k)}$. For the sake of simplicity, we dropped the second superindex $(k)$ on $z_{j,l}$, $l \in \{i, i+1\}$ here. Then, $1 - \mu_{z_{j,i}}^{z_{j+1,i},(k)}$ is the probability to recover from disease state $z_{j,i}^{(k)}$. Let further $T_{z_{j,i}}^{(k)}$ be the time in days an individual stays in a compartment $z_{j,i}^{(k)}$. For $z_j$ referring to a disease state with partial or improved immunity, we introduce the simplified factor $\kappa$ as the counterpart of the relative reduction $1 - \kappa$ between $T_{z_{j,i}^{(k)}}^{(k)}$ and the corresponding time spent in the naive compartment, e.g., for nonsymptomatic individuals

$$T_{I_{NS,II,i}}^{(k)} = \kappa\, T_{I_{NS,N,i}}^{(k)}. \tag{13}$$

Additionally, we define reduction factors $p_*$ for the extended protection against severe courses in the partial and improved immunity layers. The basic functioning of these

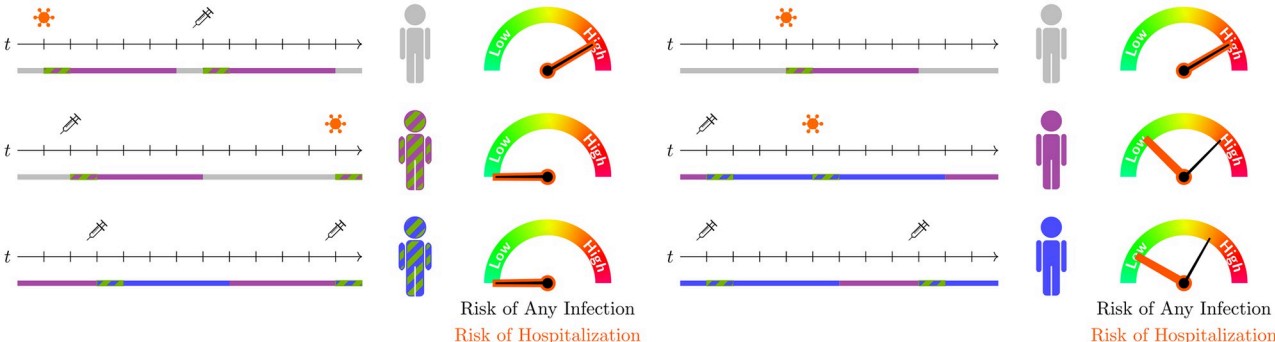

**Fig 8. Motivation on different immunity layers and waning immunity waning paces on the individual layer.** Recent infections or vaccinations protect against transmission or any infection for a short time. Less-recent immune boosters may not adequately protect against transmission but may result in good protection against a severe course of the disease.

parameters can be described by

$$\mu_{z_{j,M,i}}^{z_{j+1,M,i},(k)} = \frac{p_{z_{j+1,M,i}}}{p_{z_{j,M,i}}} \mu_{z_{j,N,i}}^{z_{j+1,N,i},(k)}, \tag{14}$$

where $i \in \{1, \ldots, G\}$ again denotes to the age group and $z_{j,*} \in \{I_{NS,*}, I_{Sy,*}, I_{Sev,*}, I_{Cr,*}\}$ refers to the disease states. Here, we added the index $M \in \{PI, II\}$ describing the affiliation to the immunity layer. Summarized, we use reduction factors to reduce the probabilities, which we define for the naive immunity layer, to use them with two other layers. For more details about the reduction factors, we refer to [26]. Furthermore, we define $T_{W_{PI},i}^{(k)}$ and $T_{W_{II},i}^{(k)}$ as the time spent in susceptible compartment $S_{PI,i}^{(k)}$ and $S_{II,i}^{(k)}$, respectively, if no infection event occurs, i.e., these times define the denominator of the waning rates. The (local) vaccination rates of the different immunity layers $M \in \{N, PI, II\}$ are given by $v_{M,i}^{(k)}$, $i \in \{1, \ldots, G\}$. Eventually, we define by $\xi_{I_{NS,i}}^{(k)}$ and $\xi_{I_{Sy,i}}^{(k)}$, $i \in \{1, \ldots, G\}$, the share of (locally) nonisolated nonsymptomatic and symptomatic infectious individuals.

In this section, we now provide the elementary model processes in a general representation. For the full model equations, see appendix S1 Text. As the susceptible compartments contain the transmission processes to the exposed compartment as well as the waning immunity from the temporary immunity compartments and to the susceptible compartment of the inferior protection layer, we provide these equations first. For the three different groups of susceptibles $M \in \{N, PI, II\}$, $i \in \{1, \ldots, G\}$, these write

$$
\begin{aligned}
\frac{dS_{M,i}^{(k)}}{dt} &= -S_{M,i}^{(k)} \rho_{M,i}^{(k)} \sum_{j=1}^{n} \phi_{i,j}^{(k)} \\
&\quad \frac{\xi_{I_{NS},j}^{(k)}(I_{NS,N,j}^{(k)} + I_{NS,PI,j}^{(k)} + I_{NS,II,j}^{(k)}) + \xi_{I_{Sy},j}^{(k)}(I_{Sy,N,j}^{(k)} + I_{Sy,PI,j}^{(k)} + I_{Sy,II,j}^{(k)})}{N_j^{D^\perp,(k)}} \\
&\quad -v_{M,i}^{(k)} S_{M,i}^{(k)} \underbrace{- \frac{1}{T_{W_M,i}^{(k)}} S_{M,i}^{(k)}}_{=:0 \text{ for } M=N} + \underbrace{\frac{1}{T_{W_{M+1},i}^{(k)}} S_{M+1,i}^{(k)}}_{=:0 \text{ for } M=II} + \underbrace{\frac{1}{T_{\mathcal{I}_M,i}^{(k)}} \mathcal{I}_{M,i}^{(k)}}_{=:0 \text{ for } M=N},
\end{aligned}
\tag{15}
$$

Here, we used the simplified notation of $M + 1 = PI$ if $M = N$ and $M + 1 = II$ if $M = PI$ and for the sake of a shorter presentation, we skipped the dependence on $t$. Furthermore, $\rho_{M,i}^{(k)}$ is the transmission risk per contact which is modeled through a baseline transmission risk $\rho_{0,N,i}^{(k)}$, a potential reduction factor for better-protected individuals $p_{E,PI}$ and $p_{E,II}$ and a parametric trignometric curve

$$\rho_{M,i}^{(k)} = \rho_{0,M,i}^{(k)} \left( 1 + k \sin\left( \pi \left( \frac{t}{182.5} + \frac{1}{2} \right) \right) \right), \quad M \in N, PI, II \tag{16}$$

with $\rho_{0,PI,i}^{(k)} = p_{E,PI} \rho_{0,N,i}^{(k)}$ and $\rho_{0,II,i}^{(k)} = p_{E,II} \rho_{0,N,i}^{(k)}$ and where $t$ is the day of the year.

Furthermore, except for the exposed and dead compartments, we can either recover from a particular disease state to a temporary immunity state or move on to an aggravated state of the disease. For any given disease state $z_j \in \mathcal{Z}_I$ excluding the different exposed states, we can

generically write

$$\frac{dz_{j,i}^{(k)}(t)}{dt} = \frac{\mu_{z_{j-1,i}}^{z_{j,i},(k)}}{T_{z_{j-1,i}}^{(k)}} z_{j-1,i}^{(k)}(t) - \frac{z_{j,i}^{(k)}(t)}{T_{z_{j,i}}^{(k)}}, \tag{17}$$

where we assume that $z_{j-1,i}$ is the previous milder infection state. For future developments and in order to determine the most influential parameters with respect to different the model outcomes, we performed a sensitivity analysis using the Morris screening method [43]. This method calculates elementary effects for each parameter by systematically varying one parameter at a time while leaving other parameters unchanged. We present the elementary effects of the individual parameters with respect to new infections and number of deaths.

In short, the sensitivity analysis indicates that the transmission risk, followed by the protection factors against any or symptomatic infection, the reduction of the infectious time for protected individuals and the overall infectious times have the highest influence on the number of new infections. The high influence of the transmission risk suggests that small changes in the probability of transmission upon contact can lead to significant variations in number of daily new infections. It should also be mentioned that a change in the probability of infection due to commutativity is equivalent to a change in the contact rate and therefore the result also applies to the latter. In terms of ICU occupancy, the analysis indicates that the probabilities of transition from the symptomatic to the severe state and from the severe state to the critical state have the highest impact, shortly followed by the transmission risk. For more details, we refer to appendix S1 Fig.

In the full model equations, we also provide confirmed compartments. However, there is no flow from undetected to detected compartments in the model equations. Detection within a node is modeled implicitly via $\xi_{I_{NS,i}}$ and $\xi_{I_{Sy,i}}$. The detected compartments are only used and filled via testing on traveling and commuting. For more details, see [34]. However, this mechanism is not used in the underlying study.

An explanation of all parameters for defining the model in Eqs (15)–(17) is provided in Table 2. In order to solve the local models, we use an adaptive Runge-Kutta Cash Karp 5(4) integration scheme [37] to ensure a small discretization error.

## Parameterization

We will showcase our novel model with a synthetic outbreak scenario and according to the developments of SARS-CoV-2 in autumn 2022 in Germany. As reported case data is divided into the six age groups: 0–4 years, 5–14 years, 15–34 years, 35–59 years, 60–79 years, and 80 years and older, we stratify our model accordingly.

**Mobility and contact patterns.**   The dataset user for inter-county commuting in Germany is based on a macroscopic transport model and a synthetic population for Germany. The transport model called DEMO [44] (short for *DEutschlandMOdell*, i.e., German Transport Model) is aimed to forecast transport in Germany when changing external factors like political policies or technological novelties. The model consists of multiple origin/destination (OD) matrices for each transport mode containing trip counts, average travel distances and average travel times for each origin/destination relation. Origins and destinations are described as traffic analysis zones. In this case the study area (Germany) is divided in 6633 traffic analysis zones which translates to matrices of size 6633x6633. The synthetic population is generated from the micro census 2017 and is spatially allocated to the traffic analysis zones. Additionally each synthetic person is binned into a behavioral homogeneous person group considering attributes like age and gender. In order to generate trip chains, e.g., home → work → leisure → shopping

**Table 2. Parameters used to define our multi-layer waning immunity model of SECIRS-type; Eqs (15)–(17).**

| Parameter | Description |
|---|---|
| $\phi_{i,j}$ | Daily contact rate between two age groups $i$ and $j$. |
| $\rho_{0,N}$ | Baseline transmission risk for people located in the naive susceptible compartments |
| $k$ | Seasonality parameter |
| $N_i^{D^\perp}$ | People from age group $i$ which have not died during simulation. |
| $\mu_{z_1}^{z_2}$ | Probability of transition from compartment $z_1$ to $z_2$. |
| $T_{z_1}$ | Time in days an individual stays in a compartment $z_1$. |
| $\xi_{I_{NS,i}}$ | Proportion of asymptomatic infectious people who are not isolated. |
| $\xi_{I_{Sy,i}}$ | Proportion of symptomatic infectious people who are not isolated. |
| $\kappa$ | Reduction factor for time spans of asymptomatic and symptomatic infections of individuals with partial or improved immunity. |
| $p_{E_{PI}}$ | Effectiveness of partial immunity protection against infection. |
| $p_{I_{Sy,PI}}$ | Effectiveness of partial immunity protection against symptomatic infection. |
| $p_{I_{Sev,PI}}$ | Effectiveness of partial immunity protection against hospitalization. |
| $p_{I_{Cr,PI}}$ | Effectiveness of partial immunity protection against ICU treatment. |
| $p_{D_{PI}}$ | Effectiveness of partial immunity protection against death. |
| $p_{E_{II}}$ | Effectiveness of improved immunity protection against infection. |
| $p_{I_{Sy,II}}$ | Effectiveness of improved immunity protection against symptomatic infection. |
| $p_{I_{Sev,II}}$ | Effectiveness of improved immunity protection against hospitalization. |
| $p_{I_{Cr,II}}$ | Effectiveness of improved immunity protection against ICU treatment. |
| $p_{D_{II}}$ | Effectiveness of improved immunity protection against death. |
| $T_{W_{PI}}$ | Rate of waning immunity of susceptible who are located in the partial immunity state. |
| $T_{W_{II}}$ | Rate of waning immunity of susceptible who are located in the improved immunity state. |
| $\nu_N$ | Vaccination rate of people with naive immunity. |
| $\nu_{PI}$ | Vaccination rate of susceptible with partial immunity. |
| $\nu_{II}$ | Vaccination rate of people with improved immunity. |

$\rightarrow$ home, daily routines including schedules, average travel distances and times are extracted from the national household survey Mobility in Germany 2017 [45]. Counts of each routine are transformed into discrete probability distributions for each person group; for a more detailed description on the process, see [46]. Further disaggregation iterates through each person where a daily routine is picked out of the according distribution. Since routines start and end at home, the start and end zone for the first respectively the last activity is known. For all activities in between, possible destination zones are selected based on travel distance from the starting zone and the trip count from the macroscopic model. The output is a list of trips between zones for each person. Finally the data is aggregated again to an OD relation level with trip counts, transport mode and person group distribution. The preliminary age-group related OD matrices have been published freely in [47]. Note that in this study age group related traffic has been recomputed by (12).

For the underlying study with an aggregated mean-value model, we only allow one travel per day and thus take workplace-related commuting from the described data set. We assume that all trips are possible in the time line of one day each. For the given data, we provide the ratio of internal and inbound commuters in Fig 9 (left) for each county in Germany, sorted in ascending order of the internal commuter ratios. In Fig 9 (right), we present the ratio of inbound commuters to the total local population of the considered county. The figure shows,

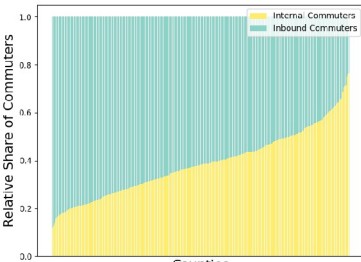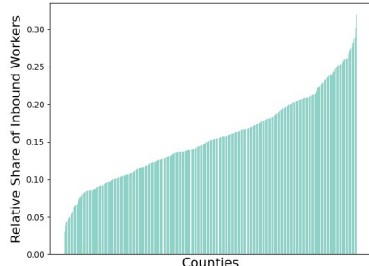

**Fig 9. Analysis of workplace commuting mobility in Germany.** Ratio of internal commuters to inbound commuters (left) and incoming commuters against the respective local population (right). The counties are sorted in ascending order.

on the one hand, the influence of inbound commuters on the number of total workers and that the share compared to the total population is relevant, if not even substantial.

**Local contact patterns.** In the previous sections, we have already explained how we model contact patterns in mobility and nonmobility settings, given survey numbers of contact $\phi_{tr}$ and $\phi_{nt}$. The contact rate is an elementary part of the calculation of the transmission or infection rate of the persons in the susceptible compartments. As in [30], we select contact locations *Home*, *Work*, *School*, and *Other* and choose our baseline contact patterns based on [48] and [49]. To explicitly model, the contacts in transportation, we additionally subdivide the group of *Other*. This is done based on the contact surveys in [50], which numbers have been provided with [51] in an accessible format. From this data we infer the relative share of contacts in transportation (with respect to other), to apply these factors element-wise to the contact matrix from [48]. Inter- and extrapolation of the aggregated contact data to the given age groups is done with recent demographic data for Germany; [30]. The final contact baseline is shown in Fig 10. Local contact reduction through NPIs is modeled as in [30]—selecting a zero minimum contact pattern.

Especially for the school contact location, vacation times are important. We have thus implemented the federal state dependent vacation times by adjusting the (school) contact matrix in the corresponding periods. We admit that our model is not (yet) adapted for, e.g., computing the import risk from hot spots outside Germany and that additional models (and

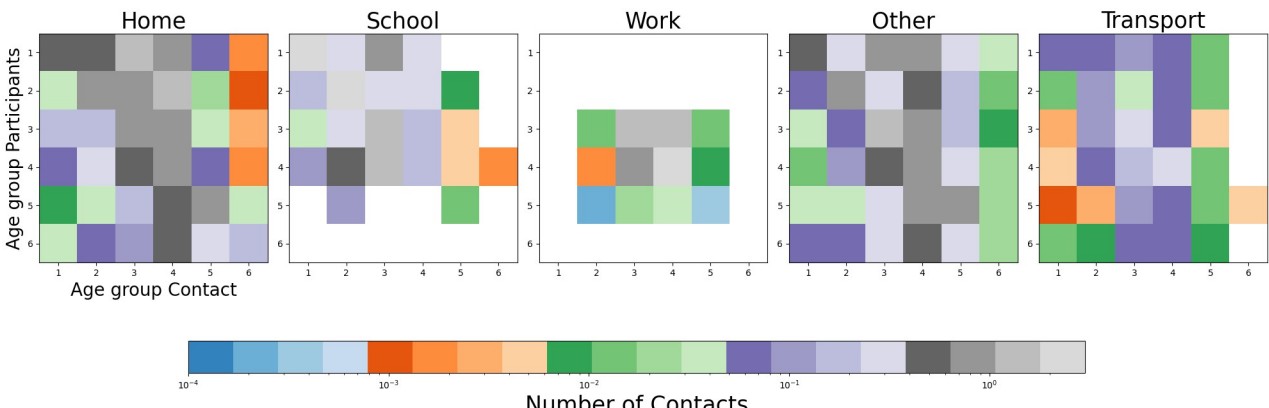

**Fig 10. Age-resolved contact matrices for Germany.** The images show the average number of contacts in each of our five contact location. Values that are zero are displayed in white.

data) might be needed for vacation time. This however is not the main focus of our improvement and could be addressed with other established models such as [52]. On average, however, as reported in previous publications [13, 30], the parametric trigonometric curve implemented for seasonality fits very well the observed surges and declines of infections over the year.

**Epidemiological parameters for SARS-CoV-2 Omicron variant BA.1/2.** In the last part of materials and methods, we focus on the epidemiological parameters specific to SARS-CoV-2 omicron variant BA.1/2, our demonstration case. The introduced model is a highly detailed model. We have explained the required parameters in Table 2. Depending on the available data, we provide stratified parameter ranges. Parameters have been obtained from an extensive literature research and prior findings with our previous models in [26, 30, 34]. Given the literature findings on waning immunity as mentioned in the introduction of the waning immunity model, we implement a rather quick immune waning for any transmission, over several months, and use $T_{\mathcal{I}_{PI}} = T_{\mathcal{I}_{II}} = 60$ days. To ensure a long-term protection against severe infection, we set $T_{W_{PI}} = T_{W_{II}} = 720$ days, which means that very little waning happens in a one year time period. For a summary see Table 3. The parameters for the baseline transmission probability and the risk of being admitted to ICU were manually fitted with the observed data, starting from the findings in [30]. Theoretically, the transmission probability $\rho$ could be chosen differently for the mobility model and the local contact model to account for varying interaction qualities. However, due to the aggregated view of ODE-based models and limited data availability, we decided to use the same value across both models, thereby assuming an identical transmission probability in both models. For most parameters, we provide a range from which we uniformly take parameter samples. By making a large number of Monte Carlo runs, we obtain good estimate of the uncertainty in the output.

To determine the initial states for the individual compartments of our model, we use the German case numbers published by the Robert Koch-Institute [68]. The idea of transforming the confirmed case data to initial states for each compartment was given in [26]. We follow this approach while implementing the changed immunity layers in this approach. For more details, we refer to appendix S1 Text. The presented SECIRS model is highly parameterized. We therefore use a sensitivity analysis to determine the parameters that have the greatest impact on the disease dynamics.

## Results

In this section, we use and validate the introduced SECIRS-type model with three different immunity layers and two paces for waning immunity and the novel travel-time aware mobility model.

We will consider several settings in corresponding subsections:

- First, we consider the separated effects of the waning immunity model, validated qualitatively over a two years time frame with wastewater data.

- Second, we numerically validate that the novel mobility method is indeed a generalization of [30] and convergences to the old method if maximum travel time goes to zero.

- Third, we provide a synthetic outbreak scenario with a infection hotspot in Cologne and show how infection numbers spread differently with the newly introduced method.

- Fourth, we consider the Post-Oktoberfest and beginning of winter season, where we validate our model based on reported ICU occupancy.

- Finally, we provide a retrospective analysis of SARS-CoV-2 transmissions in the late phase of the pandemic and also use ICU occupancy as additional validation.

**Table 3. Model parameters for spatially resolved model and Omicron BA.1/2.** If parameters are stratified by age, vertical lines separate the corresponding columns. Column *Reference* provides used references in addition to the findings in [26, 30, 34].

| Param. | 0–4 | 5–14 | 15–34 | 35–59 | 60–79 | 80+ | Reference |
|---|---|---|---|---|---|---|---|
| | | | **Age Group** | | | | |
| $\rho_{0,N}$ | [0.03, 0.06] | [0.075, 0.105] | | | [0.12, 0.15] | [0.15, 0.225] | [30] & Fitted |
| $k$ | [0.1,0.3] | | | | | | [13, 26, 30] |
| $\xi_{I_{NS}}$ | sigmoidal curve from 0.5 to 1 on incidence 10 to 20 | | | | | | [34] |
| $\xi_{I_{Sy}}$ | sigmoidal curve from [0.0, 0.2] to [0.4, 0.5] on incidence 10 to 150 | | | | | | [34] |
| $T_E$ | 1.66 | | | | | | [53–55] |
| $T_{I_{NS}}$ | 1.44 | | | | | | [53, 55] |
| $T_{I_{Sy}}$ | [6.58,7.16] | | | | | | [56] |
| $T_{I_{Sev}}$ | [1.8,2.3] | | [2.5,3.67] | | [3.5,5] | [4.91,7.01] | [57] |
| $T_{I_{Cr}}$ | [9.29, 10.57] | | [10.842,12.86] | | [11.15, 13.23] | [11.07,13.25] | [57] |
| $T_{\mathcal{I}_{PI}}$ | 60 | | | | | | [39, 58] |
| $T_{\mathcal{I}_{II}}$ | 60 | | | | | | [39, 58] |
| $T_{W_{PI}}$ | 720 | | | | | | [39, 58] |
| $T_{W_{II}}$ | 720 | | | | | | [39, 58] |
| $\mu_{I_{NS}}^{I_{Sy}}$ | [0.60,0.80] | | [0.7,8.3] | | | [0.85,0.9] | [59] |
| $\mu_{I_{Sy,N}}^{I_{Sev,N}}$ | [0.006,0.009] | [0.0048,0.0072] | [0.006,0.0092] | [0.00147,0.0222] | [0.0375,0.045] | [0.07,0.0875] | [28, 60, 61] |
| $\mu_{I_{Sev,N}}^{I_{Cr,N}}$ | [0.0104,0.0208] | [0.0104,0.0208] | [0.0104,0.0208] | [0.0208,0.0416] | [0.052,0.0728] | [0.0728,0.0936] | [30] & Fitted |
| $\mu_{I_{Cr,N}}^{D_N}$ | [0.00,0.039] | | [0.039,0.0702] | | [0.117,0.195] | [0.195,0.273] | [62–64] |
| $\kappa$ | 0.5 | | | | | | [65] |
| $p_{E_{PI}}$ | 1.0 | | | | | | [39] |
| $p_{I_{Sy,PI}}$ | [0.039, 0.254] | | | | | | [66] |
| $p_{I_{Sev,PI}}$ | [0.18, 0.48] | | | | | | [67] |
| $p_{I_{Cr,PI}}$ | see $p_{I_{Sev,PI}}$ | | | | | | |
| $p_{D_{PI}}$ | see $p_{I_{Sev,PI}}$ | | | | | | |
| $p_{E_{II}}$ | 1.0 | | | | | | [39] |
| $p_{I_{Sy,II}}$ | [0.656, 0.705] | | | | | | [66] |
| $p_{I_{Sev,II}}$ | [0.81, 0.9] | | | | | | [67] |
| $p_{I_{Cr,II}}$ | see $p_{I_{Sev,II}}$ | | | | | | |
| $p_{D_{II}}$ | see $p_{I_{Sev,II}}$ | | | | | | |

## Validation of a global waning immunity model using wastewater data

Given the complexity of our proposed model, it is important to show the advancements on the particular elements separately. Therefore, we consider the results of the immune-waning model separately from the impact of the mobility scheme and use a single (waning immunity) model for the whole of Germany over a two years time frame (2022–2024). The parameters used in this analysis differ slightly from those used in the spatially resolved model (Tables 3 and 4) since we homogeneously aggregate heterogeneous immunity dynamics over the whole of Germany. We prevented from tweaking parameters too much to improve the global fit, as inferred parameters would necessarily be wrong on a local scale. As changing or temporarily absent testing strategies lead to highly biased numbers of confirmed cases, we use wastewater data as a less biased proxy to adequately capture disease dynamics. From Fig 11, we see that the new model qualitatively represents the three major surges and declines over the two year

**Table 4. Model parameters for the global SECIRS-Type model without spatial resolution.** The table shows the basic transmission probability ($\rho_i^{(0)}$) and the increased protection ($p_{E_{PI}}$ and $p_{E_{II}}$) for individuals with higher immunity.

| Parameter | Age Group | | | | | |
|---|---|---|---|---|---|---|
| | **0–4** | **5–14** | **15–34** | **35–59** | **60–79** | **80+** |
| $\rho_{0,N}$ | [0.06, 0.12] | [0.15, 0.21] | | | [0.24, 0.3] | [0.3, 0.4] |
| $p_{E_{PI}}$ | 0.4 | | | | | |
| $p_{E_{II}}$ | 0.4 | | | | | |

period indicated by the wastewater samples from the AMELAG [69]. On the other hand, the old model wrongly results in an eradication of the virus in the population. We furthermore see that the development of reported cases diverges significantly from the wastewater measurements. We see that the global model does not fit well several local peaks induced by a heterogeneous spread of COVID-19 in Germany, see also the local wastewater measurements in appendix S2 Fig. Additional considerations to use and validate our novel model in different scenarios will be the focus of future work.

## Numerical validation of novel mobility scheme

In the section "Theoretical properties of the novel mobility model", we have already shown that the novel mobility scheme reduces to the previously suggested model [30], as the travel time approaches zero. For validation, we here present the numerical convergence of the novel model to the previously proposed model. To simplify disease dynamics, we use local SIR models (1) for the five German counties Cologne, Leverkusen, Rheinisch-Bergischer Kreis, Rhein-

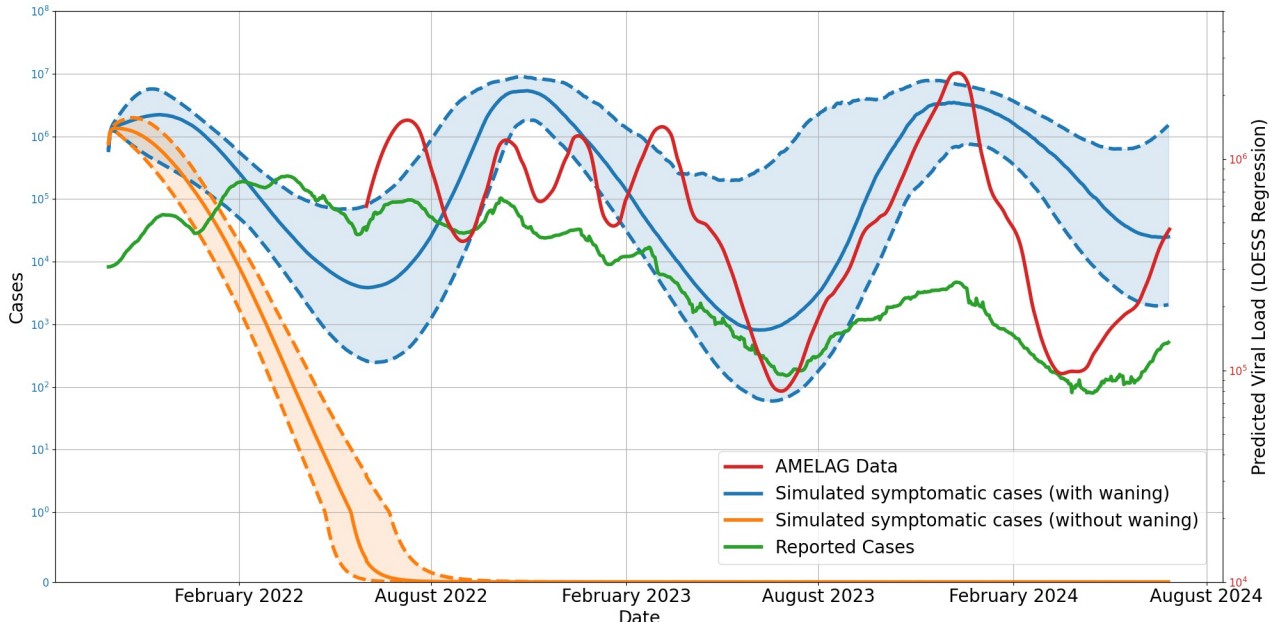

**Fig 11. Comparison of the simulated and reported cases with wastewater measurements over a two-year period (2022–2024).** New symptomatic infected individuals simulated with novel immune-waning model (blue) and existent no-waning model (orange). Obtained AMELAG wastewater data (red) and reported cases (green). The solid line represents the median of the simulation results, while the shaded area indicates the 25th to 75th percentile.

**Table 5. Numerical convergence of the novel mobility scheme.** The table shows the normed difference between the outcomes of the previous and the novel mobility scheme after 50 days of simulation with values of $t_{tr}^{max} \rightarrow 0$.

| $t_{tr}^{max}$ | Difference | | | | | |
|---|---|---|---|---|---|---|
| | *Total* | *Cologne* | *Leverkusen* | *Rhein.-Berg. Kreis* | *Rhein-Sieg-Kreis* | *Bonn* |
| $1.00e + 00$ | $1.10e + 05$ | $4.27e + 04$ | $1.18e + 04$ | $1.48e + 04$ | $2.46e + 04$ | $1.61e + 04$ |
| $1.00e - 01$ | $7.81e + 03$ | $3.12e + 03$ | $8.69e + 02$ | $6.12e + 02$ | $1.73e + 03$ | $1.48e + 03$ |
| $1.00e - 02$ | $7.58e + 02$ | $2.92e + 02$ | $9.18e + 01$ | $1.05e + 02$ | $1.69e + 02$ | $1.01e + 02$ |
| $1.00e - 03$ | $7.56e + 01$ | $2.91e + 01$ | $9.15e + 00$ | $1.04e + 01$ | $1.69e + 01$ | $1.01e + 01$ |
| $1.00e - 04$ | $7.55e + 00$ | $2.90e + 00$ | $9.14e - 01$ | $1.04e + 00$ | $1.69e + 00$ | $1.01e + 00$ |
| $1.00e - 05$ | $7.55e - 01$ | $2.90e - 01$ | $9.14e - 02$ | $1.04e - 01$ | $1.69e - 01$ | $1.01e - 01$ |
| $1.00e - 06$ | $7.55e - 02$ | $2.90e - 02$ | $9.14e - 03$ | $1.04e - 02$ | $1.69e - 02$ | $1.01e - 02$ |
| $1.00e - 07$ | $7.55e - 03$ | $2.90e - 03$ | $9.14e - 04$ | $1.04e - 03$ | $1.69e - 03$ | $1.01e - 03$ |
| $1.00e - 08$ | $7.55e - 04$ | $2.90e - 04$ | $9.14e - 05$ | $1.04e - 04$ | $1.69e - 04$ | $1.01e - 04$ |
| $1.00e - 09$ | $7.55e - 05$ | $2.90e - 05$ | $9.14e - 06$ | $1.04e - 05$ | $1.69e - 05$ | $1.01e - 05$ |
| $1.00e - 10$ | $7.55e - 06$ | $2.90e - 06$ | $9.14e - 07$ | $1.04e - 06$ | $1.69e - 06$ | $1.01e - 06$ |
| $1.00e - 11$ | $7.56e - 07$ | $2.91e - 07$ | $9.11e - 08$ | $1.05e - 07$ | $1.68e - 07$ | $1.01e - 07$ |

Sieg-Kreis, and Bonn with realistic demographies and parameters as given in Fig 2. For the initial conditions, we set 0.1% of the total populations to be infected and zero recovered individuals. We then compute the solution over 50 days using the novel mobility scheme and subsequently evaluate the difference to the solution obtained with the existing instantaneous travel mobility scheme [30] at the last date. Table 5 shows that, as travel time decreases, the difference between the two model outputs converges to zero. Additionally, we present the convergence behavior using map plots for the different values of $t_{tr}^{max}$ in Fig 12.

## Fictional outbreak scenario

In this section, we want to show the impact of the mobility on the evolution of the disease using a fictional outbreak scenario. Therefore, we assume that in the initial stage to only have a single county which has a high amount of exposed people. In this context, we assume 5% of Cologne's population to be exposed to the virus. Although this number is immense, it was chosen by intention to directly visualize the qualitative propagation of new cases to connected counties.

To emphasize the influence of mobility, we compared our results with the mobility scheme presented in [30] (here denoted *existing instantaneous travel method*, both times using our multi-layer waning immunity model. The advantage of the novel mobility approach accounting for travel and transmission during commuting, contrasting the immediate transfer of travelers in [30].

In the maps of Fig 13, we illustrate the progression of the total number of currently infected individuals using both approaches over 90 days (excluding Cologne to restrict the colorbar). In particular in the left-most part of the figure, we see that the virus attains counties which have not or almost not been affected with the existing instantaneous travelmethod. We see that the novel approach is leading to a different propagation pattern also easily attaining indirect or loosely coupled regions. The very large number of exposed in Cologne leads to the rapid increase in the upper right part. The second and higher peak in October is due to the waning against any transmission and our seasonality factor in the transmission processes. If we look at the number of infections in all counties except Cologne, a comparison of the median values of the two methods shows that far more individuals will get infected using the novel approach.

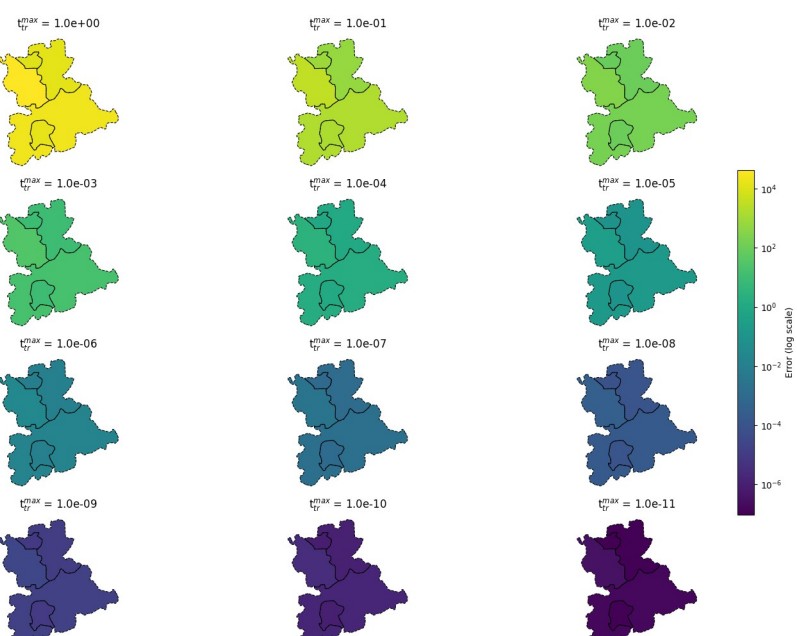

**Fig 12. Numerical convergence of novel mobility scheme on map plots.** The figure presents a grid of maps, each illustrating the spatial distribution of normed difference as provided in Table 5 for the respective county. The counties displayed are Cologne, Leverkusen, Rheinisch-Bergischer Kreis, Rhein-Sieg-Kreis, and Bonn. The maps use a logarithmically scaled colorbar. Using geodata "Verwaltungsgebiete 1:2 500 000, Stand 01.01. (VG2500)" from https://gdz.bkg.bund.de, copyright; GeoBasis-DE / BKG 2021, license dl-de/by-2-0, see https://www.govdata.de/dl-de/by-2-0.

This is in accordance with the exponential increase in case number in so far virus-free regions. Similar process had been observed for the beginning of the pandemic when only a small number of regions was affected at all. The corresponding process is also reflected in the plots of the maps, where we already see a much more divided infection pattern on the first day.

## Developments of Post-Oktoberfest until winter season 2022 in Bavaria

To further validate the effect of the novel mobility model, we simulate the Post-Munich Oktoberfest period until the beginning of the winter season 2022. The Oktoberfest, which took place from September 17 to October 3, 2022 and attracted around 5.7 million visitors [70]. Unfortunately, conducted tests have decreased significantly over 2022 and positive rate has been high, see [71] and Fig 14, which makes reported case numbers quite unreliable and comparisons difficult. On an aggregated level we thus also include reported ICU numbers from [72]. We initialize the model based on the official reported data one week after the festival, at October 10 2022, and simulate until the end of the year using the parameters in Table 3. We adjusted the transmission probability $\rho$ with a factor of 0.8. While the simulate mild cases largely exceed the reported cases (see Fig 14 (left)), which is no surprise given the low number of tests and high positive rate, the simulated ICU admissions come close to the reported ICU admissions; see Fig 14 (right).

To provide spatial details of our simulations, we present the simulated number of symptomatic infections in a spatially resolved plot of Bavaria. In Fig 15, we see the spread of infections across the federal state, with the numbers presented relative to the local population.

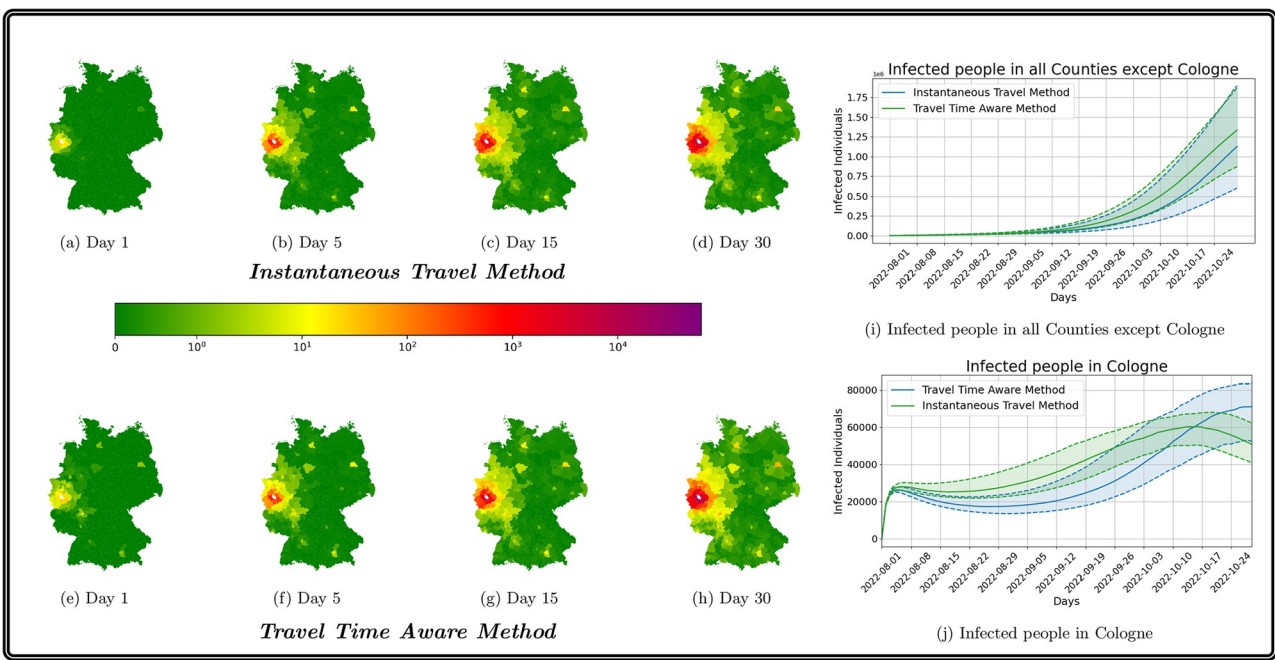

**Fig 13. Propagation of infections with existing instantaneous travel method [30] and novel travel time aware method for an outbreak scenario.** Median propagation of infections in German counties (except Cologne) over 90 days with the existing method (top row maps) and the novel method (bottom row maps). In the plots of the right hand side, we provide the evolution of case numbers in Cologne and the rest of Germany. The shaded area visualizes the p25 and p75 percentile results of the 500 ensemble runs. The median value is given as solid line. Using geodata "Verwaltungsgebiete 1:2 500 000, Stand 01.01. (VG2500)" from https://gdz.bkg.bund.de, copyright; GeoBasis-DE / BKG 2021, license dl-de/by-2-0, see https://www.govdata.de/dl-de/by-2-0.

## Late-phase pandemic in Germany

We now consider the late-phase SARS-CoV-2 pandemic in Germany. We begin our simulation on August 1st, 2022 and examine the official face mask obligation in public transport, which was maintained until February 2023 and allowed both FFP2 and surgical masks [73]. Beginning October 1, 2022, nationwide enforcement of the use of FFP2 masks on long-distance public transportation [73] was implemented. However, there were differences in local public transportation between the German federal states. In Lower Saxony [74] and Hamburg [75] the FFP2 mask obligations also existed in local public transport but were dropped at the start of October 2022.

The newly introduced mobility scheme offers us a flexible approach to implement actions within the mobility node, without necessitating changes to the underlying local model. Within our model, compliance with the mask mandate can be represented by altering the effective contact rate through a damping of the contacts. As shown in [76], the proper usage of FFP2 masks is crucial. For both individuals correctly wearing FFP2 masks, the risk of infection during a 20-minute encounter with an infected person was estimated to be as small as 0.1%. Notably, [76] also demonstrates that without any form of mask-wearing, the probability of infection approaches 100%. Therefore, we implement the effect of FFP2 masks by a reduction factor of 0.1% to the effective contact.

Throughout much of the pandemic, the German national testing strategy was predominantly focused on testing individuals displaying symptoms [77]. Therefore, we assume that the number of confirmed cases corresponds directly to the daily growth of symptomatic

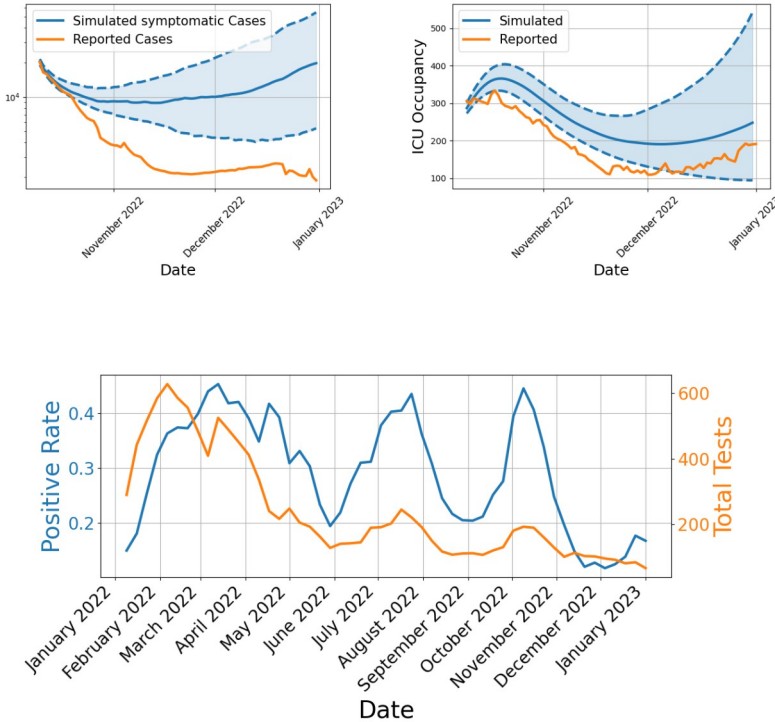

**Fig 14. Comparison of simulated and reported data for Bavaria and conducted tests.** The top left figure shows simulated daily new symptomatic against reported cases in Bavaria, with data smoothed using a 7-day moving average to reduce oscillations. The top right figure compares simulated ICU admissions to reported ICU data. The bottom figure presents the total weekly number of tests conducted per 100 thousand individuals alongside with the positive test rate in Bavaria for the entire year 2022.

individuals if the detection rate stays constant. This is also in line with the overall decreasing accuracy in positive-testing asymptomatic cases with regard to the Omicron variant [78]. As official reporting was suspected to have large underdetection factors, we based our comparisons not only on the official reported case numbers but also on the self-reported cases in the

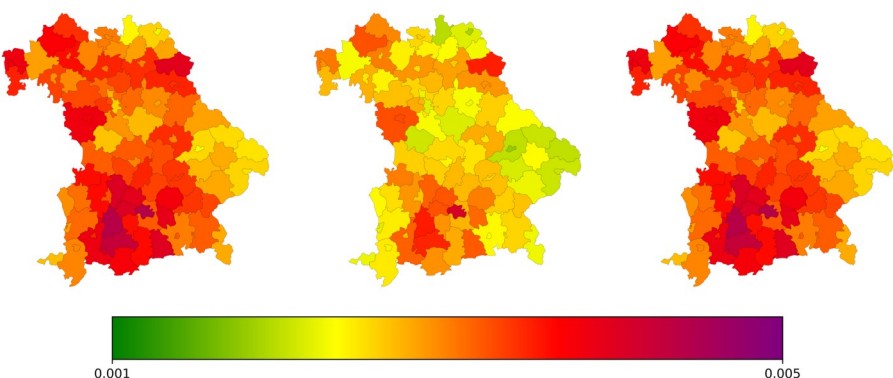

**Fig 15. Spatial distribution of simulated symptomatic infections in Bavaria for selected days of the Post-Oktoberfest and beginning winter season 2022.** Presented values are provided relative to local population. Using geodata "Verwaltungsgebiete 1:2 500 000, Stand 01.01. (VG2500)" from https://gdz.bkg.bund.de, copyright; GeoBasis-DE / BKG 2021, license dl-de/by-2-0, see https://www.govdata.de/dl-de/dl-de/by-2-0".

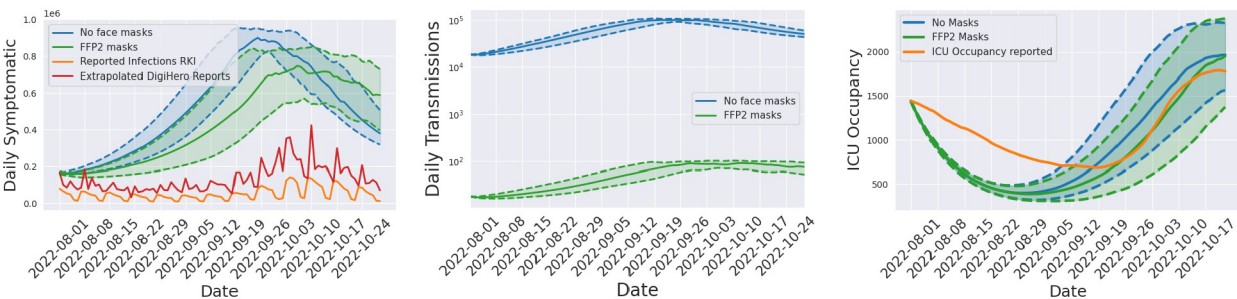

**Fig 16. Daily number of symptomatic infections, transmissions and ICU occupancy.** The simulations provide results using correctly worn FFP2 masks (green) and no face masks (blue) in the mobility models. The number of daily symptomatic infections (left, blue and green) is compared to data provided by the DigiHero study (red), where the number of participants is extrapolated to the population in Germany and the official reported cases (yellow). Median values are indicated by the solid lines and shaded areas between the dotted lines represent the p25 and p75 percentiles of the particular ensemble runs. In the center plot, we visualize the daily transmission in the mobility models during the course of the simulation on a logarithmic scale. Furthermore, simulated ICU occupancy with different mask usage (right, blue and green) is compared to reported ICU occupancy (right, orange).

DigiHero [79] study. We extrapolated the reported infections from 33730 participants in autumn 2022 to the total population of Germany. In Fig 16 (left), we provide the results of two different ensemble runs with 500 individual runs each where either no face masks (blue) or FFP2 (green) masks in transportation have been simulated. We visualize shaded areas for the regions between the p25 and p75 percentiles of the 500 runs and solid lines for the median results.

Until beginning of October, Fig 16 (left) shows substantially larger numbers of infected individuals for the scenario without face masks. The quick decline in numbers in this scenario is due to the very large number of infected individuals before and the temporal immunity saturation obtained. We see that face masks in transportation alone cannot prevent peaking of case numbers. However, the overall peak is considerably smaller (around 20%). Face masks in transportation can thus contribute in ensuring functionality of critical infrastructure and preventing hospital bed shortages. Compared to reported data, we still see a nonnegligible difference. However, we also assume important underdetection, as test-positive rates range between 30 and 50%; see [80]. In all curves, we similarly see a first stagnation in numbers and decline of infections towards end of the considered time frame.

An important advantage of our approach lies in its ability to quantify infections occurring during participation in traffic, achieved through the utilization of the traffic models. This offers a fundamental advantage over conventional structures. The significantly different probabilities of infection due to the implementation of a mask obligation in the transport models are clearly reflected in the daily number of transmissions in the traffic models obtained by our simulation; see Fig 16 (center). In order to validate the results obtained by our simulation, we compare to ICU occupancy in Germany from [72] which is a more stable and reliable indicator. In Fig 16 (right), we see that our model is able to capture the reported ICU occupancy for the second part of the simulation very well while discharges on initially admitted are overestimated. Unfortunately, we could not resolve the mismatch in the beginning of the simulation period as ICU data is reported without age distributions, while average time spent on ICU varies with age [30, 57]. We here tried to extrapolate a demographic distribution of initial ICU occupancy using demographic distributions of reported cases. In future works, more advanced methods for initial state estimation will be developed.

Finally, we want to illustrate the spread of the disease on a geographical basis. In Fig 17, we provide the (median) daily new numbers of symptomatic infected individuals on county level.

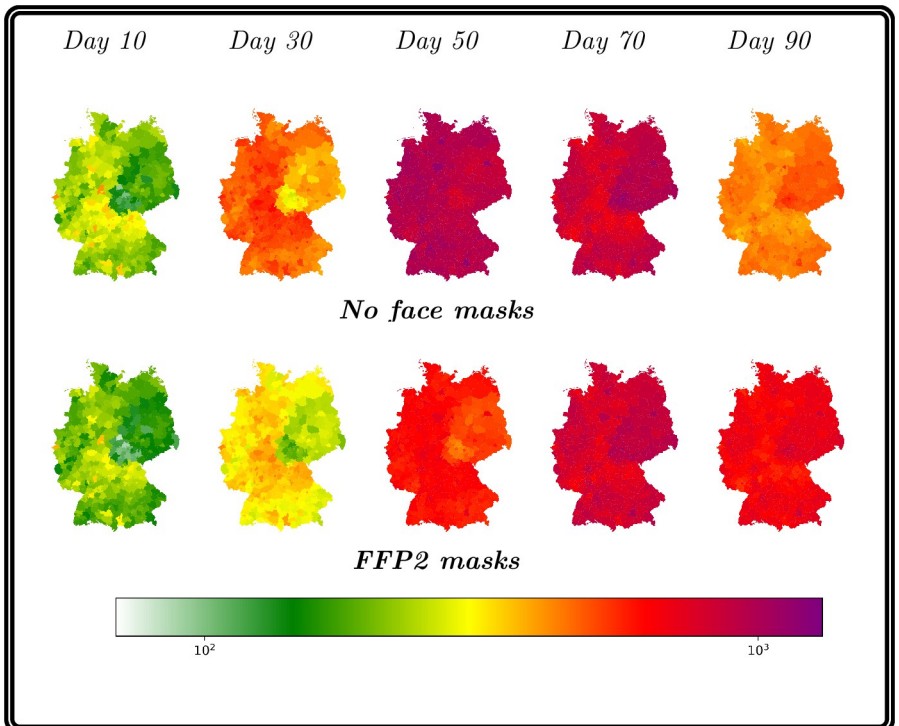

**Fig 17. Daily new number of symptomatic infected individuals on county level.** This figure illustrates the change in the daily number of symptomatic infected German counties over the course of 90 days beginning with the 1st August 2022. Each subfigure displays a different day with the two different modes (No masks or FFP2 masks). Using geodata "Verwaltungsgebiete 1:2 500 000, Stand 01.01. (VG2500)" from https://gdz.bkg.bund.de, copyright; GeoBasis-DE / BKG 2021, license dl-de/by-2-0, see https://www.govdata.de/dl-de/by-2-0.

This clearly emphasizes once again that the masks, as used only in the mobility models, have a significant influence on the spread of infection. The far higher peak in the case where no face masks are worn is clearly recognizable around day 50. Again, due to a temporal immunity saturation, we find a decline from day 70 to day 90 in the nonrestricted scenario while the face mask scenario numbers decline slower.

## Discussion

The results presented in this study are based on a mean-value model, a multitude of assumptions and different parameters. Although based on an extensive literature review and previous findings in [26, 30, 34], there is still uncertainty about many (virus-specific) parameters. Although the distinction into 3–4 different immunity layers through seropositivity studies [81, 82], a division into three different subpopulations is a rough aggregation. Furthermore, our model is not suited for simulation the evolution of virus variants or the competing circulation of different variants. An inherent limitation of almost all mean-field models and most contact surveys is the difficulty to address (potential) transmission through aerosols which cannot (or only to limited extent) be measured in (direct) contacts so that estimated parameters have to address for this limitation. However, mathematical models can help to explore the mechanisms in infectious disease spread and reasonable assumptions are necessary, even for models up to digital twins. We have waived a lot of unrealistic assumptions such as homogeneous mixture in space and in demography. Our model has been resolved for meaningful regions and

demography has been taken into account to model different vulnerabilities for different age groups. In order to better assess model performance under changing parameters, we have included a sensitivity analysis on the parameters used in the waning immunity model in the appendix. The provided quantification for the different parameter effects will guide extending or reducing the model in future works and analyses.

As we stated ourselves, a critical factor in infectious disease modeling is mobility. Since the ground truth for daily mobility is difficult to grasp, again, assumptions have to be used. We have used the output of a state-of-the-art macroscopic mobility model to be used in our simulations. Different mobility patterns clearly lead to different virus distribution patterns. Furthermore, we have focused on daily mobility activities in this study. The seasonal impact of, e.g., vacation times on travel behavior were not taken into account and influences from abroad have not been included here. For the summer vacation time, where travel activities significantly change with travel from hot or to hot spot regions (outside Germany), our model might need enrichment by an import risk model to reliably capture disease dynamics from the outside.

A precise quantification of waning immunity against different courses of the disease given particular immunization histories is still an open research question. To best cope with this problem, we have considered two large and recent review articles from which we derived most reasonable parameters. However, there is still large uncertainty in several positions which we cannot reduce. Qualitatively, our model could correctly predict stagnation, decline and resurges in considered settings. The effect of waning immunity to SARS-CoV-2 over more than 1 or 2 years, the interplay with new variants or vaccinations is still an open research question our model cannot answer yet.

In order to get meaningful predictions for future developments, our model needs to initialized with reasonable values of the population's immunity. As the reporting of vaccinations is not conducted on personal level and as the dark figure of unreported transmission numbers increased with the end of the pandemic, the current status of the population is difficult to obtain. Serological studies may be needed to provide approximated starting values for future predictions. These limitations in data and assumptions should be kept in mind when interpreting the results of this work.

## Conclusion

In this paper, we have presented a novel travel-time aware metapopulation model combined with a novel multi-layer waning immunity model based on ordinary differential equations. Both models advance already existing models for infectious disease dynamics by taking into account yet neglected but essential properties for virus propagation.

The travel-time aware metapopulation model advances existing metapopulation or hybrid graph-ODE approaches by considering travel time as well as transmissions during transport. As unconstrained mobility clearly is a driver for infectious disease dynamics, more realistic mobility modeling increases the reliability of model outcomes. The travel-time aware metapopulation model can directly be combined with any kind of ODE-based transmission model and thus enhances global predictions by correctly resolving heterogeneously different spreading dynamics.

The novel multi-layer waning immunity model copes with the problem of heterogeneous and hybrid immunity in a population in mid- or late-phase pandemics as well as in endemic scenarios. The novel model does not only provide three different populations with different protection factors but also two different paces of waning immunity, i.e., against any transmission and against a severe course of the disease—a property that has been observed for

SARS-CoV-2. Given reparameterization, it can also be used for other human-to-human transmittable diseases that show similar developments.

With the combination of both models, we could correctly assess developments in the late phase of the pandemic in Germany in 2022. Although vaccination could already mitigate the number of severe courses of the diseases, large numbers of mild cases can also have a disastrous impact on economy and critical infrastructure. Our model serves as a good basis for future waves SARS-CoV-2 in upcoming autumn or winter seasons and adaptation to other viruses is possible. The suggested mobility model can be of great help when particular interventions in public transportation are envisaged and expected transmissions can be assessed beforehand. As it can be combined with any kind of ODE-based model, it can also be used for other diseases such as Influenza or RSV.

## Supporting information

**S1 Text. Supplementary tables, figures, and analyses.** Fig A. Initialization of the Exposed compartment. For an arbitrary but fixed $t_0$, individuals who got exposed in the left, red area will get symptoms or recover in the right, blue area. Table A. Parameters used to define our multi-layer waning immunity model of SECIRS-type.
(PDF)

**S1 Fig. Results of the sensitivity analysis showing the elementary effects for the model parameters with respect to the daily new infections (left) and ICU occupancy (right).** The impact of each parameter is represented by a logarithmic scaled bar where the length represents the positive and negative influence.
(PDF)

**S2 Fig. Recorded AMELAG wastewater data by federal state.** Predicted viral load (LOESS regression) per sample location and sorted by federal state as obtained by [69].
(PDF)

## Author Contributions

**Conceptualization:** Henrik Zunker, David Kerkmann, Martin J. Kühn.

**Data curation:** Henrik Zunker, Alain Schengen, Sophie Diexer, Martin J. Kühn.

**Formal analysis:** Henrik Zunker, David Kerkmann, Martin J. Kühn.

**Funding acquisition:** Rafael Mikolajczyk, Michael Meyer-Hermann, Martin J. Kühn.

**Investigation:** Henrik Zunker, David Kerkmann, Alain Schengen, Martin J. Kühn.

**Methodology:** Henrik Zunker, David Kerkmann, Martin J. Kühn.

**Project administration:** Martin J. Kühn.

**Resources:** Rafael Mikolajczyk, Michael Meyer-Hermann, Martin J. Kühn.

**Software:** Henrik Zunker, René Schmieding.

**Supervision:** Michael Meyer-Hermann, Martin J. Kühn.

**Validation:** Henrik Zunker, René Schmieding, David Kerkmann, Alain Schengen, Sophie Diexer, Rafael Mikolajczyk, Michael Meyer-Hermann, Martin J. Kühn.

**Visualization:** Henrik Zunker, Martin J. Kühn.

**Writing – original draft:** Henrik Zunker, Martin J. Kühn.

**Writing – review & editing:** Henrik Zunker, René Schmieding, David Kerkmann, Alain Schengen, Sophie Diexer, Rafael Mikolajczyk, Michael Meyer-Hermann, Martin J. Kühn.

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
