## [Decision Letter · Decision Letter 0]

27 Jun 2024

Dear Dr. Kühn,

Thank you very much for submitting your manuscript "Novel travel time aware metapopulation models: A combination with multi-layer waning immunity to assess late-phase epidemic and endemic scenarios" for consideration at PLOS Computational Biology.

As with all papers reviewed by the journal, your manuscript was reviewed by members of the editorial board and by several independent reviewers. In light of the reviews (below this email), we would like to invite the resubmission of a significantly-revised version that takes into account the reviewers' comments.

Associate editor

The paper has been read by two experts in the field, and to a lesser extent by me. All of us find the paper potentially interesting but having some issues needing attention. Both referees make many important points, the overall most critical point probably being the fitting and evaluation of the model given its complexity. Please address all questions/comment raised by the referees. I would also consider modifying the current title which is quite long and not very informative.

Kind regards, Tom Britton

We cannot make any decision about publication until we have seen the revised manuscript and your response to the reviewers' comments. Your revised manuscript is also likely to be sent to reviewers for further evaluation.

Sincerely,

Tom Britton

Academic Editor

PLOS Computational Biology

Virginia Pitzer

Section Editor

PLOS Computational Biology

Associate editor

The paper has been read by two experts in the field, and to a lesser extent by me. All of us find the paper potentially interesting but having some issues needing attention. Both referees make many important points, the overall most critical point probably being the fitting and evaluation of the model given its complexity. Please address all questions/comment raised by the referees. I would also consider modifying the current title which is quite long and not very informative.

Kind regards, Tom Britton

Reviewer's Responses to Questions

**Comments to the Authors:**

Reviewer #1: The review is uploaded as an attachment.

Reviewer #2: The authors present a new epidemiological model that introduces novelty in two fronts: a/ travel time in a graph-ODE metapop model (including passing explicitly through multiple districts), b/ a multi-layer waning immunity model that support different paces for protection against various degrees of disease severity. The authors combine these advancements to investigate the impact of mobility (more specifically the use of mitigation measures such as masks during mobility) in a late-phase epidemic scenario.

Overall, I believe the authors present interesting modelling frameworks, that cover important aspects related to the modelling of infectious diseases. I studied the proposed frameworks which appears to be sound, and I could not find any immediate errors or mistakes. However, such a complex model requires many choices, both regarding model structures and the selected parameters. I think it would be appropriate to discuss the limitations and uncertainties regarding these choices properly in the discussion section, which currently seems to be quite limited.

My main remarks regarding this paper concern the evaluation of the model frameworks, the presentation, some assumptions made about the impact of mitigation strategies while travelling, and reproducibility.

The authors present important advances regarding metapop modelling (time-aware and passing through multiple districts) and immunity waning. However, while these components are not necessarily connected, they are evaluated in combination, which makes it hard to assess how these components match their individual goals. I would expect an evaluation on the SECIRS compartment model and an evaluation that compares the graph-ODE model from [Kuhn, 2021] with the methodological extensions presented in this paper (so, without the SECIRS model). In my opinion the comparison between the graph-ODE and the method presented in this paper can be performed on an early-phase pandemic scenario, when waning is not a concern.

Next, I appreciate the combined setting of a late-phase epidemic, which is interesting. However, at this point, I find it hard to assess how this model is able to capture reality. I tried to make this assessment by interpreting Figure 11, which shows the green model curve (which should be close to the German reality during the time frame that was modelled, if I'm not mistaken) and 2 data curves (red and orange, where red corresponds to the DigiHero reports). Based on this Figure, I find it hard to see how well the red curve and the green curve match. Perhaps a rescaling of the DigiHero curve could help here, but perhaps it would be better to try to capture the DigiHero reporting dynamics in the model to allow for curves that are easier to compare? That being said, I understand that comparing symptomatic curves is not an easy thing to do (due to underreporting, as mentioned by the authors), in that regard, did the authors consider taking hospitalisations into account for evaluating the models (I would argue hospitalisation data has proven more stable during the course of the course of the pandemic)?

Regarding the presentation of the methods. It was not always easy to identify the authors' contributions given that the background and the novel methods are quite intertwined. I think it would be good to explicitly emphasise the contributions and add some more structure to the method section to help the reader. Perhaps it could help to split this section in a background section and a methods section, to make this more clear?

Unless I missed it, the authors consider the type of travel contacts the same as contacts made at home, work, school. I would argue that contacts made during a travel (on the train/bus etc) are more distant, as individuals might not talk to each other. Earlier work in the context of influenza distinguished between conversational and physical contacts (https://journals.plos.org/ploscompbiol/article?id=10.1371/journal.pcbi.1002425), but the types of contacts made at public transport might be considered to be of a different type? Can the authors comment on this, as assumptions in this regard could have an impact on some of the conclusions (e.g., Fig 11).

Regarding reproducibility, the authors state "The total number of produced simulation data sums up to 200 GB and cannot be uploaded easily.". I would argue that the simulation results are not that important, but the code and data sources needed to reproduce these results are key. Can the authors explain whether this is possible with the material that is available on GitHub?

Some minor remarks:

- although fighting infectious diseases -> sounds strange, consider rephrase

- situations of high transmission (risk) -> situations of high transmission

- Over the last years, to predict SARS-CoV-2 development in 19 Germany, contributions have been made by a variety of different approaches. -> Rephrase: Over the last years, contributions have been made by a variety of different approaches, to predict SARS-CoV-2 development in 19 Germany."

- In the introduction, the authors provide a list of different types of models. I have two comments. First, I would split between agent-based and individual-based models. Second, the authors only seem to mention German models, but I think for the individual-based models, models like COVASIM and "STRIDE COVID-19" should also be mentioned? Same remark for the other model categories.

- Mathematical models based on systems of ordinary differential equations (ODE) -> Mathematical models based on systems of ODEs (abbreviation was already introduced)

- When first reading through equation (1), I was confused that there was a patch notation (k), which it concerns a definition that does not consider a metapop approach yet. I think it would be more clear to first introduce a normal SIR model.

- "although the model is a pure infection dynamics model, just parameterized for mobility settings." -> During the first read, this was hard to understand, consider a rephrase.

- "while only a portion p(k) tr ∈ [0, 1] also has contacts in traffic locations (which is nontrivial as people might not" -> don't see why this is nontrivial, I believe it is trivial that only a portion has contacts during travelling? Or do I misunderstand the point of the authors?

- we need to determine who is commuting at all. -> not clear, rephrase

- A important challenge

- where neither infection nor vaccination promises lasting protection -> where neither infection nor vaccination *establishes* lasting protection

- "larger numbers of infected" -> "larger numbers of infected individuals" (this way of writing is also used in the rest of the manuscript, so I would consider changing it there as well)

- The authors indicate that one variable was fitted (Table 3), but I did not find any additional explanation, unless I missed it.

- number is ridiculously large, -> rephrase

- (here denoted Existing method, -> missing a closing )

- "Existing method" -> perhaps a more descriptive name of the original model would be more clear

References:

Kuhn MJ, Abele D, Mitra T, Koslow W, Abedi M, Rack K, et al. Assessment of effective mitigation and prediction of the spread of SARS-CoV-2 in Germany using demographic information and spatial resolution. Mathematical Biosciences. 2021;

**Have the authors made all data and (if applicable) computational code underlying the findings in their manuscript fully available?**

Reviewer #1: None

Reviewer #2: **No: **I chose "No" because this was not clear yet to me. I address this in my detailed comments. Based on the feedback of the authors I will reconsider this choice.

PLOS authors have the option to publish the peer review history of their article (what does this mean?). If published, this will include your full peer review and any attached files.

Reviewer #1: No

Reviewer #2: No
---

## [Decision Letter · Decision Letter 1]

21 Oct 2024

Dear Dr. Kühn,

Thank you very much for submitting your manuscript "Novel travel time aware metapopulation models and multi-layer waning immunity for late-phase epidemic and endemic scenarios" for consideration at PLOS Computational Biology. As with all papers reviewed by the journal, your manuscript was reviewed by members of the editorial board and by several independent reviewers. The reviewers appreciated the attention to an important topic. Based on the reviews, we are likely to accept this manuscript for publication, providing that you modify the manuscript according to the review recommendations.

Both reviewers were happy with the revision, and so am I. Reviewer 2 just have some very minor issues which you should be able to deal with quickly.

Kind regards, Tom Britton, Academic editor

Sincerely,

Tom Britton

Academic Editor

PLOS Computational Biology

Virginia Pitzer

Section Editor

PLOS Computational Biology

Both reviewers were happy with the revision, and so am I. Reviewer 2 just have some very minor issues which you should be able to deal with quickly.

Kind regards, Tom Britton, Academic editor

Reviewer's Responses to Questions

**Comments to the Authors:**

Reviewer #1: All comments were addressed. Thank you!

Reviewer #2: I thank the authors for a very thorough and detailed revision, and congratulate them on this nice work.

This revision addresses my comments and I only have a couple of minor remarks:

- "We use wastewater data as an unbiased estimator" -> A bit strange to refer to data as an estimator? Moreover, I would not state that it is unbiased, cause that would be hard to show (also not convinced that it is). Perhaps rephrase as "We use wastewater as a proxy"?

- In Figure 4, the authors state "Spatial distribution of symptomatic infections in Bavaria". Given the revision that was done, perhaps it is better to show ICU cases her as well? And if this adjustment is made, perhaps give some information on how well these predictions follow the reported ICU data from a spatially perspective?

- The authors state "In Fig. 5 (right), we see that our model is able to capture the reported ICU occupancy for the second part of the simulation very well while discharges on initially admitted are overestimated." Can the authors add some discussion on the mismatch (reported vs model) between initially admitted ICU cases.

**Have the authors made all data and (if applicable) computational code underlying the findings in their manuscript fully available?**

Reviewer #1: None

Reviewer #2: Yes

PLOS authors have the option to publish the peer review history of their article (what does this mean?). If published, this will include your full peer review and any attached files.

Reviewer #1: **Yes: **Mohamed El Khalifi

Reviewer #2: No

Figure Files:

Data Requirements:

Reproducibility:

References:

---

## [Decision Letter · Decision Letter 2]

12 Nov 2024

Dear Dr. Kühn,

We are pleased to inform you that your manuscript 'Novel travel time aware metapopulation models and multi-layer waning immunity for late-phase epidemic and endemic scenarios' has been provisionally accepted for publication in PLOS Computational Biology.

Best regards,

Tom Britton

Academic Editor

PLOS Computational Biology

Virginia Pitzer

Section Editor

PLOS Computational Biology

Feilim Mac Gabhann

Editor-in-Chief

PLOS Computational Biology

Jason Papin

Editor-in-Chief

PLOS Computational Biology

Now also the 2nd reviewer is satisfied with the revision, and so am I. I am hence happy to propose that the manuscript is accepted for publication.

Kind regards, Tom Britton

Reviewer's Responses to Questions

**Comments to the Authors:**

Reviewer #2: I thank the authors for their reply and this nice manuscript. I have no more comments. A final suggestion is that a short summary of their response to the second question could be added to the manuscript, as this might also interest the readers.

**Have the authors made all data and (if applicable) computational code underlying the findings in their manuscript fully available?**

Reviewer #2: Yes

PLOS authors have the option to publish the peer review history of their article (what does this mean?). If published, this will include your full peer review and any attached files.

Reviewer #2: No

---

## [Editor Report · Acceptance letter]

9 Dec 2024

PCOMPBIOL-D-24-00368R2 

Novel travel time aware metapopulation models and multi-layer waning immunity for late-phase epidemic and endemic scenarios

Dear Dr Kühn,

I am pleased to inform you that your manuscript has been formally accepted for publication in PLOS Computational Biology. Your manuscript is now with our production department and you will be notified of the publication date in due course.

With kind regards,

Anita Estes
